# The genetic landscape of basal ganglia and implications for common brain disorders

Shahram Bahrami [1,2,15] ✉, Kaja Nordengen [1,3,15], Jaroslav Rokicki[4], Alexey A. Shadrin[1,2], Zillur Rahman[2], Olav B. Smeland [1], Piotr P. Jaholkowski[1], Nadine Parker [1], Pravesh Parekh [1], Kevin S. O'Connell[1], Torbjørn Elvsåshagen [1,3,5], Mathias Toft[1,3], Srdjan Djurovic [1,6], Anders M. Dale [7,8,9,10], Lars T. Westlye [1,11], Tobias Kaufmann [1,12,13] & Ole A. Andreassen [1,2,14] ✉

The basal ganglia are subcortical brain structures involved in motor control, cognition, and emotion regulation. We conducted univariate and multivariate genome-wide association analyses (GWAS) to explore the genetic architecture of basal ganglia volumes using brain scans obtained from 34,794 Europeans with replication in 4,808 white and generalization in 5,220 non-white Europeans. Our multivariate GWAS identified 72 genetic loci associated with basal ganglia volumes with a replication rate of 55.6% at P < 0.05 and 87.5% showed the same direction, revealing a distributed genetic architecture across basal ganglia structures. Of these, 50 loci were novel, including exonic regions of *APOE*, *NBR1* and *HLAA*. We examined the genetic overlap between basal ganglia volumes and several neurological and psychiatric disorders. The strongest genetic overlap was between basal ganglia and Parkinson's disease, as supported by robust LD-score regression-based genetic correlations. Mendelian randomization indicated genetic liability to larger striatal volume as potentially causal for Parkinson's disease, in addition to a suggestive causal effect of greater genetic liability to Alzheimer's disease on smaller accumbens. Functional analyses implicated neurogenesis, neuron differentiation and development in basal ganglia volumes. These results enhance our understanding of the genetic architecture and molecular associations of basal ganglia structure and their role in brain disorders.

[1]Institute of Clinical Medicine, University of Oslo, Oslo, Norway. [2]KG Jebsen Centre for Neurodevelopmental disorders, University of Oslo, Oslo, Norway. [3]Department of Neurology, Oslo University Hospital, Oslo, Norway. [4]Centre of Research and Education in Forensic Psychiatry, Oslo University Hospital, Oslo, Norway. [5]Department of Behavioral Medicine, Institute of Basic Medical Sciences, University of Oslo, Oslo, Norway. [6]Department of Medical Genetics, Oslo University Hospital, Oslo, Norway. [7]Multimodal Imaging Laboratory, University of California San Diego, La Jolla, CA, USA. [8]Department of Psychiatry, University of California, San Diego, La Jolla, CA, USA. [9]Department of Neurosciences, University of California San Diego, La Jolla, CA, USA. [10]Department of Radiology, University of California, San Diego, La Jolla, CA, USA. [11]Department of Psychology, Faculty of Social Sciences, University of Oslo, Oslo, Norway. [12]Department of Psychiatry and Psychotherapy, Tübingen Center for Mental Health, University of Tübingen, Tübingen, Germany. [13]German Center for Mental Health (DZPG), Tübingen, Germany. [14]Department of Psychiatry, Oslo University Hospital, Oslo, Norway. [15]These authors contributed equally: Shahram Bahrami, Kaja Nordengen. ✉e-mail: shahram.bahrami@medisin.uio.no; ole.andreassen@medisin.uio.no

The basal ganglia are a group of interconnected subcortical nuclei deep in the brain[1]. The major parts of the basal ganglia are located in the cerebrum and include the caudate nucleus, putamen, globus pallidus and the accumbens area[2,3]. Substantia nigra in the midbrain and the subthalamic nucleus in diencephalon may be regarded as associated structures. Although the basal ganglia comprise physically distinct entities, the individual nuclei exhibit a robust functional cohesion due to their intricate interconnections, and is often regarded as a functional unit[4].

The basal ganglia integrate and modulate cortical information, and are involved in motor[5], cognitive[6] and limbic functions[3,7]. The functional unit is involved in motor functions through initiation, execution, and coordination of movements[8], action selection[6] and in learning and execution of procedural memory and habits, allowing actions to become automatic and efficient over time[9,10]. Beyond motor control, the basal ganglia play important roles in cognitive functions like decision-making[6], the shifting of attention, updating information, and adapting behavior[6,11–13], in addition to reward processing, which shapes behavior, decision-making[14] and motivation[15]. The basal ganglia are involved in emotional processing and the regulation of affective states[3,7], and integration of emotional information in decision-making processes[3,16]. However, there is strong evidence supporting that the different basal ganglia are integrated, as the function of the individual nuclei depends on input from neighboring nuclei, making basal ganglia an interconnected network[4].

Likely due to their widespread connections to other parts of the brain, the basal ganglia have been implicated in several brain diseases, ranging from neurodegenerative to psychiatric and neurodevelopmental conditions. The most well-known disorder related to the basal ganglia is Parkinson's disease (PD), where loss of dopaminergic projections from the substantia nigra to the basal ganglia leads to progressive motor symptoms. In Alzheimer's disease (ALZ), evidence suggests that tau- and amyloid aggregation can affect the basal ganglia and contribute to cognitive and motor impairments[17]. Through pain processing and modulation, basal ganglia are also involved in the pathophysiology of the primary headache disorder migraine (MIG). Dysfunction within the basal ganglia circuits may contribute to stereotyped/repetitive movements, reduced attentional control[12,13] and social and emotional processing[7] often observed in individuals with autism spectrum disorder (ASD)[18,19]. Through its role in regulating motor hyperactivity[20,21], reward processing[22–24] and inhibitory control[25,26], the basal ganglia is also central for the pathophysiology of attention-deficit/hyperactivity disorder (ADHD)[25–27]. Dysregulation of dopamine neurotransmission in the mesocorticolimbic pathway involving the basal ganglia has been implicated in ADHD[28–30], but also related to hallucinations and delusions in schizophrenia (SCZ)[31,32], while cortico-striatal circuitry abnormalities may contribute to the cognitive impairments observed in SCZ[33,34]. The basal ganglia also play a role in mood regulation and emotional processing[35,36], and basal ganglia functions have been implicated in both bipolar disorder (BIP)[36] and major depressive disorder (MDD)[35,37]. Studies of basal ganglia volumes have in many circumstances showed inconsistent results. For MIG[38], PD[39], MDD[40], ASD[13], and ADHD[41,42] some studies report reduced basal ganglia volumes, while others report no significant differences[43,44], or even striatal enlargement[45–49].

While the functional aspects of the basal ganglia have been extensively studied, the genetic architecture underlying their structural characteristics remains largely unexplored. There are, however, genetic studies on volumetrics of subcortical structures which includes basal ganglia nuclei[50,51], without focusing on the nuclei as a functional unit. Hibar and colleagues[50] reported one locus significantly associated with caudate nucleus and four loci significantly associated with putamen, but none with accumbens area or globus pallidus. Satizabal and colleagues[51], on the other hand, reported four loci associated with the accumbens area, 10 loci with the caudate nucleus, six loci with globus pallidus and nine loci with putamen. A

comprehensive understanding of the genetic mechanisms shaping basal ganglia volumes is lacking. The current study represents the multivariate genome wide association study (GWAS) on the basal ganglia volumes, an analytical approach that can take advantage of basal ganglia as one functional unit. Thus, there are likely pleiotropic genetic variants shared across the different nuclei, which can be identified with the multivariate MOSTest method[52]. Still, we also include univariate GWASs for comparisons with earlier studies[50,51]. Understanding the genetic basis of basal ganglia volumes as a whole can provide insights into the pathogenesis and etiology of basal ganglia-related brain disorders, which can form the basis for the future development of targeted therapeutic interventions.

## Results

### Univariate GWASs reveal 47 novel genetic loci associated with basal ganglia

Univariate GWAS analyses on individual nuclei collectively known as the basal ganglia revealed 60 unique loci, 43 of which were novel. We identified 11 loci associated with accumbens area (Supplementary Data 1, Supplementary Fig. 1), where three out of the four loci reported by Satizabal et al.[51] are replicated, leaving eight novel loci. We report 26 genetic loci significantly associated with the caudate nucleus (Supplementary Data 1, Supplementary Fig. 1), of which 18 loci are novel, due to replication of eight out of the ten loci reported by Satizabal et al.[51], while the one locus reported by Hibar et al.[50] is not replicated. For pallidum, we report 13 significantly associated genetic loci (Supplementary Data 1, Supplementary Fig. 1), nine of which are novel, while four are replications of the six loci reported by Satizabal et al.[51]. For putamen, we replicate all four loci reported associated with putamen by Hibar et al.[50] and seven out of the nine loci reported by Satizabal et al.[51], in addition to 15 novel loci, altogether 23 loci, significantly associated with putamen in our univariate GWAS (Supplementary Data 1, Supplementary Fig. 1). We have also performed a univariate GWAS on basal ganglia as a functional unit (one volume), resulting in 22 significant loci spanning 12 chromosomes (Supplementary Data 1, Supplementary fig. 1). Of these loci, 14 have not been previously reported from the univariate GWASs from the individual nuclei of which basal ganglia consists[50,51]. There is a large overlap between the loci identified in the univariate GWASs of the different basal ganglia nuclei (mean 22%, range 9–38%) and between the univariate GWAS of the basal ganglia as one volume and univariate GWASs of the different basal ganglia nuclei (mean 42%, range 27–65%), supporting the basal ganglia as a functional group of intertwingled subcortical nuclei.

All basal ganglia nuclei showed significant single nucleotide polymorphism (SNP)-heritability (Fig. 1A), with highest estimates for the caudate ($h^2 = 0.34$, se=0.025) and the whole basal ganglia ($h^2 = 0.31$, se=0.026). Figure 1B shows a correlation matrix of basal ganglia volumes, with phenotypic correlation shown in upper-left section and genetic correlation using the univariate GWAS summary statistics shown in the lower-right section. As expected, genetic correlations mapped phenotypic correlations (Supplementary Data 2). The overall lowest genetic correlations were found between pallidum and accumbens (rg = 0.1153, see = 0.056), which are also the structures with the lowest estimated heritability.

### Multivariate GWAS reveals 72 genetic loci associated with basal ganglia

A multivariate GWAS deployed using the MOSTest framework[53] identified 72 significant independent loci, including 50 novel loci (Supplementary Data 3). We here adopt a conservative definition of novelty, wherein we characterize it as loci identified through the multivariate MOSTest approach for the basal ganglia as a unified functional unit, ensuring non-overlapping with any loci previously identified in univariate GWASs for any of the individual basal ganglia nuclei.We thresholded based on

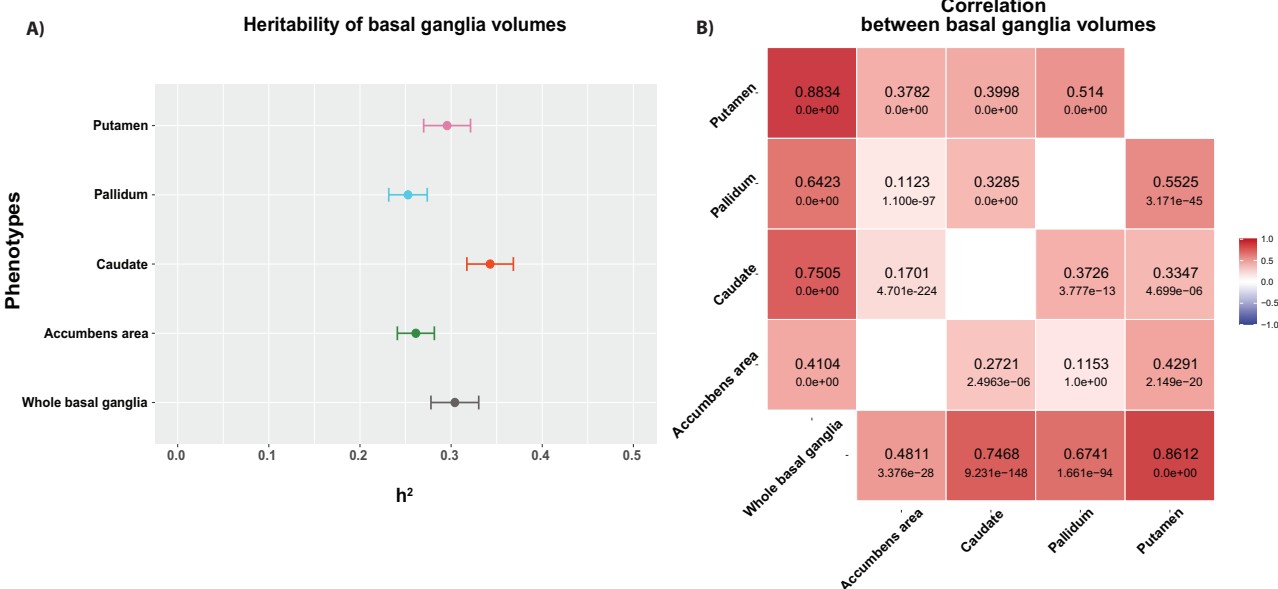

**Fig. 1 | The univariate signatures of basal ganglia volumes indicate high heritability and significant phenotypic and genetic correlations. A** Heritability of basal ganglia volumes ($n = 34,794$), where the numbers depict SNP-heritability estimates (h2). In the plot, the dots represent mean heritability for each subregion of the basal ganglia, while the error bars represent the standard error (SEM): Putamen in pink (h2 = 0.296, SEM = 0.0026), Pallidum in light blue (h2 = 0.253,

SEM = 0.021), Caudate in red (h2 = 0.342, SEM = 0.025), Accumbens area in green (h2 = 0.261, SEM = 0.02) and whole basal ganglia in gray (h2 = 0.304, SEM = 0.026). **B** LD-score regression-based genetic correlations (in lower-right section) and phenotypic correlation (in upper-left section) between each pair of regions, using the univariate GWAS summary statistics.

genome-wide significance ($P < 5 \times 10^{-8}$) and identified a total of 12305 candidate SNPs, 294 independent significant SNPs and 89 lead SNPs across structures located in the 72 genomic loci, using the FUMA platform[54] (Supplementary Data 4). Figure 2 illustrates the results of multivariate and univariate GWAS, where the upper part of the Miami plot shows the multivariate polygenic architecture across basal ganglia volumes. A distributed genetic architecture throughout the basal ganglia structure is supported by the higher multivariate statistics in comparison with the univariate statistics for various basal ganglia volumes in most of the identified loci (Fig. 1), which is further confirmed by genetic correlation analysis of the individual volumes (Supplementary Data 2).

Although the strongest associations among the 72 significant independent loci that were identified in the multivariate framework are also present in the results of univariate analysis, a considerable proportion of these loci demonstrated higher effects that were not significant at the genome-wide level. Leveraging these distributed effects across the various subregions, the multivariate approach resulted in enhanced discovery. Q–Q plots from MOSTest analysis (Supplementary Fig. 2) including one from permutation testing, showed successive validity of the multivariate test statistic with the LD-score regression intercept determined to be 1.024. The findings were supported by a multivariate replication study from white UK Biobank and a multivariate generalization (non-white) study that produced the same effect direction for 87.5% of the lead SNPs (Supplementary Fig. 3).

**Functional annotation, gene mapping and genetic analyses**
We functionally annotated SNPs associated with basal ganglia volumes that were in LD ($r^2 \geq 0.6$) with one of the independent significant SNPs with $P < 5 \times 10^{-8}$ in the discovery sample using FUMA v1.4.1[54]. A majority of these SNPs were intronic (53.5%) or intergenic (29.1%) and 1.1% were exonic (Supplementary Fig. 4A and Supplementary Data 4). Supplementary Fig. 4B provides information for functional SNP categories for the basal ganglia volume. About 81.8% of the SNPs had a minimum chromatin state of 1–7, thus suggesting they were in open chromatin regions (Supplementary Fig. 4C)[55,56]. Three of the lead SNPs were exonic and combined annotation-dependent depletion (CADD) scores of those

SNPs were 23.1 (rs13107325), 18.4 (rs2070835) and 12.64 (rs429358), thus indicating deleterious protein effects[57] (Supplementary Data 3). rs13107325 and rs429358 are located in *SLC39A8* and *APOE*, respectively, and have previously been associated with SCZ, PD and ALZ[58–61].

Genome-wide gene-based association analyses (GWGAS; $P < 2.622 \times 10^{-6}$, i.e., 0.05/19073 genes) using MAGMA v1.08[62] detected 149 unique genes across the basal ganglia (Supplementary Data 5). Supplementary Fig. 5 provides Manhattan and Q–Q plots for the GWGAS. Gene-set analyses using MAGMA identified significant Gene Ontology sets for neurogenesis, neuron differentiation and development (Supplementary Data 6).

**Open target genetics**
We additionally used Open Target Genetics to identify the target genes for each lead SNP of the 72 loci implied for the basal ganglia volumes (Fig. 3 and Supplementary Data 3).

We used the 73 uniquely mapped genes from Open Targets for gene set analysis, pathway analysis and differentially expressed genes (DEG) analysis. GO gene-set analysis for the target genes of the significant loci revealed 16 significantly associated biological processes, including 'GO_NEURON_DIFFERENTIATION', 'GO_NEUROGENESIS' and 'GO_CELL_PART_MORPHOGENESIS' (Supplementary Data 7). These genes were also significantly associated with four cellular component gene-sets, including 'GO_NEURON_PROJECTION' and 'GO_SYNAPSE' (Supplementary Data 7). There were 19 pathways significantly over-represented among the target genes of the significant loci with 'the DSCAM interactions', 'Axon guidance', 'Nervous system development' and 'Netrin-1 signaling' as the most significant (Supplementary Data 8).

We also tested the tissue specificity using the DEG sets defined for the target genes of the identified loci. The results show that these target genes are significantly expressed in the brain and specifically in basal ganglia (Supplementary Figs. 6, 7).

In addition, we found enrichment for the mapped genes in the lymphatic system, nervous systems, and sensory systems (Supplementary Fig. 8A); the top enriched cell types were mostly related to the immune and nervous systems (natural killer cells, T cells, dendritic

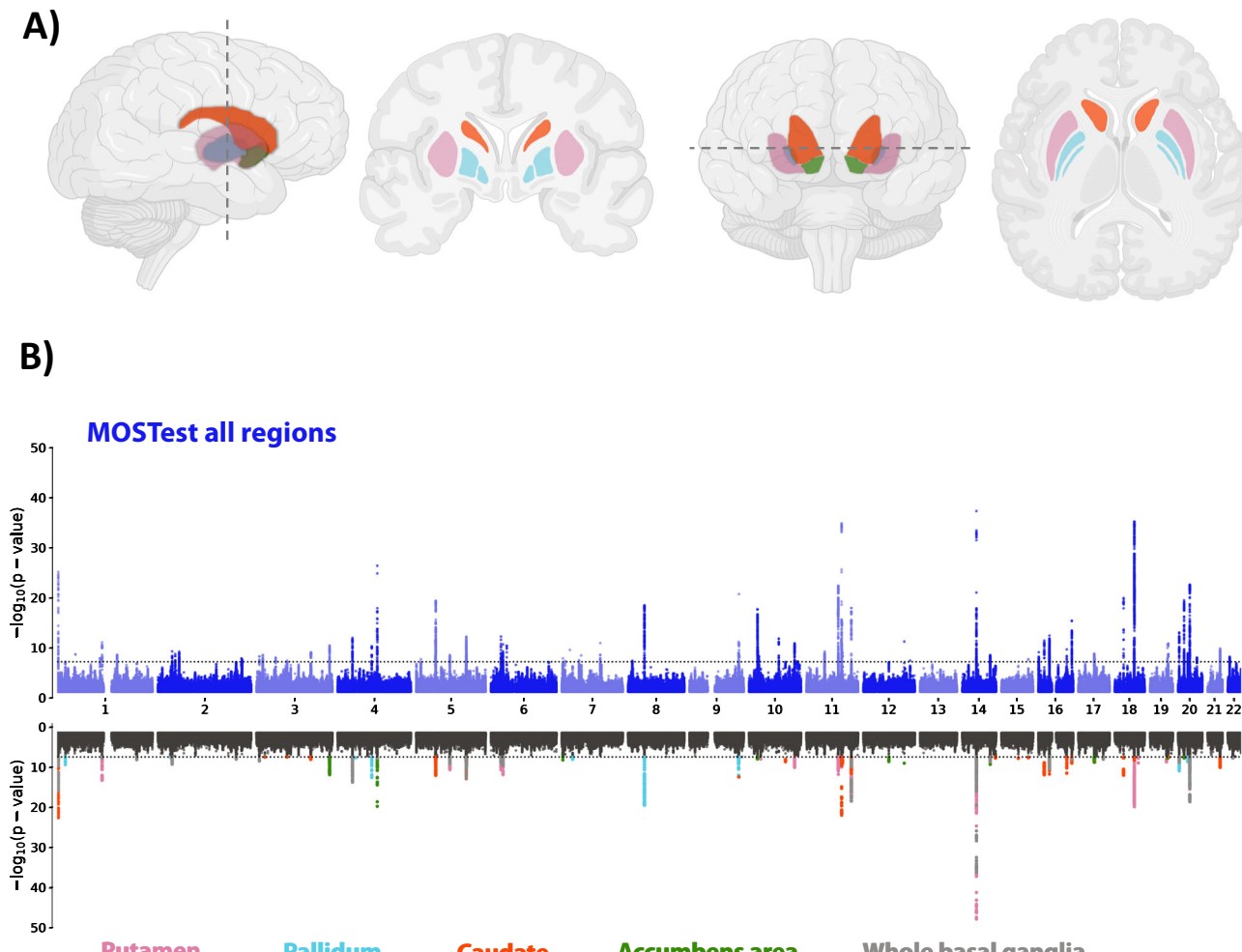

**Fig. 2 | The genetic architecture of basal ganglia. A** Schematic illustration of the basal ganglia regions, comprising the anatomically distinguishable subfields of putamen (pink), pallidum (light blue), caudate (red) and accumbens (green). Created with BioRender.com released under a Creative Commons Attribution-NonCommercial-NoDerivs 4.0 International license **B** The upper part of the Miami plot illustrates the −log$_{10}$($P$) statistic from the multivariate GWAS across the entire basal ganglia (Multivariate Omnibus Statistical Test (MOSTest)), with 72 significant loci. For comparison, the lower part depicts for each of the 72 unique loci the corresponding −log10($P$) statistics from univariate GWASs of single subregions (one color per subregion, black indicates non-significant SNPs, $p$-values are two-tailed), supporting a distributed genetic architecture across the basal ganglia structure. The color codes for the schematic illustration (**A**) and the Miami plot (**B**) are consistent and given in the bottom part of the figure.

cells, bipolar neurons, inhibitory neurons, and endothelial cells) (Supplementary Fig. 8B). We also showed that the mapped genes expressed in the brain have cell-type-specific expression patterns in the main cell types of the human cerebral cortex (Supplementary Fig. 9).

We determined protein–protein and co-expression networks for the mapped genes. The genes have 14.17 and 16.41 physical interaction and gene co-expression, respectively (Supplementary Fig. 10). A gene drug interaction analysis shows that 13 out of 73 genes have interaction with 124 drugs (Supplementary Data 9), including antipsychotic (risperidone, chlorpromazine) and antiepileptic/mood stabilizing drugs (lamotrigine, carbamazepine). Within the UK Biobank, data on drug use are available for 60 out of the 124 drugs included in the gene-drug interaction analysis. The number of individuals in our cohort using any of these 60 drugs are illustrated for relevant characterization of our cohort (Supplementary Fig. 11).

### Genetic overlap between the basal ganglia and common brain disorders

To further examine the polygenic architecture of basal ganglia volumes and the potential genetic overlap between basal ganglia and common

brain disorders, we used GWAS summary statistics for ADHD, ASD, BIP, MDD, SCZ, ALZ, MIG, and PD. Genetic correlations of the disorders with individual basal ganglia subregions revealed only a significant association with PD after Bonferroni correction (Supplementary Fig. 12 and Supplementary Data 10). Conditional Q–Q plots conditioning the multivariate statistic of basal ganglia on the disorders and vice versa clearly demonstrated a pattern of pleiotropic enrichment in both directions (Supplementary Fig. 13) as a leftward shift in the group of SNPs with higher significance. This shift towards significance in the conditional Q-Q plot implies that the genetic variants associated with one phenotype (either basal ganglia or brain disorders) are also enriched for association with the other phenotype, supporting the notion of shared genetic factors or pleiotropy, as outlined[63–65]. Conjunctional FDR analysis allowed us to test for shared loci between the basal ganglia and each of the disorders. We identified 3 loci significantly overlapping with ADHD, 2 loci with ASD, 20 with BIP, 83 with SCZ, 15 with MDD, 33 with MIG, 21 with ALZ and 28 with PD (Fig. 4 and Supplementary Data 11-18).

### Gene mapping and overlapping shared significant loci and genes
A full list of loci overlapping between basal ganglia and the eight disorders is provided in Supplementary Datas 11–18. Figure 5 illustrates

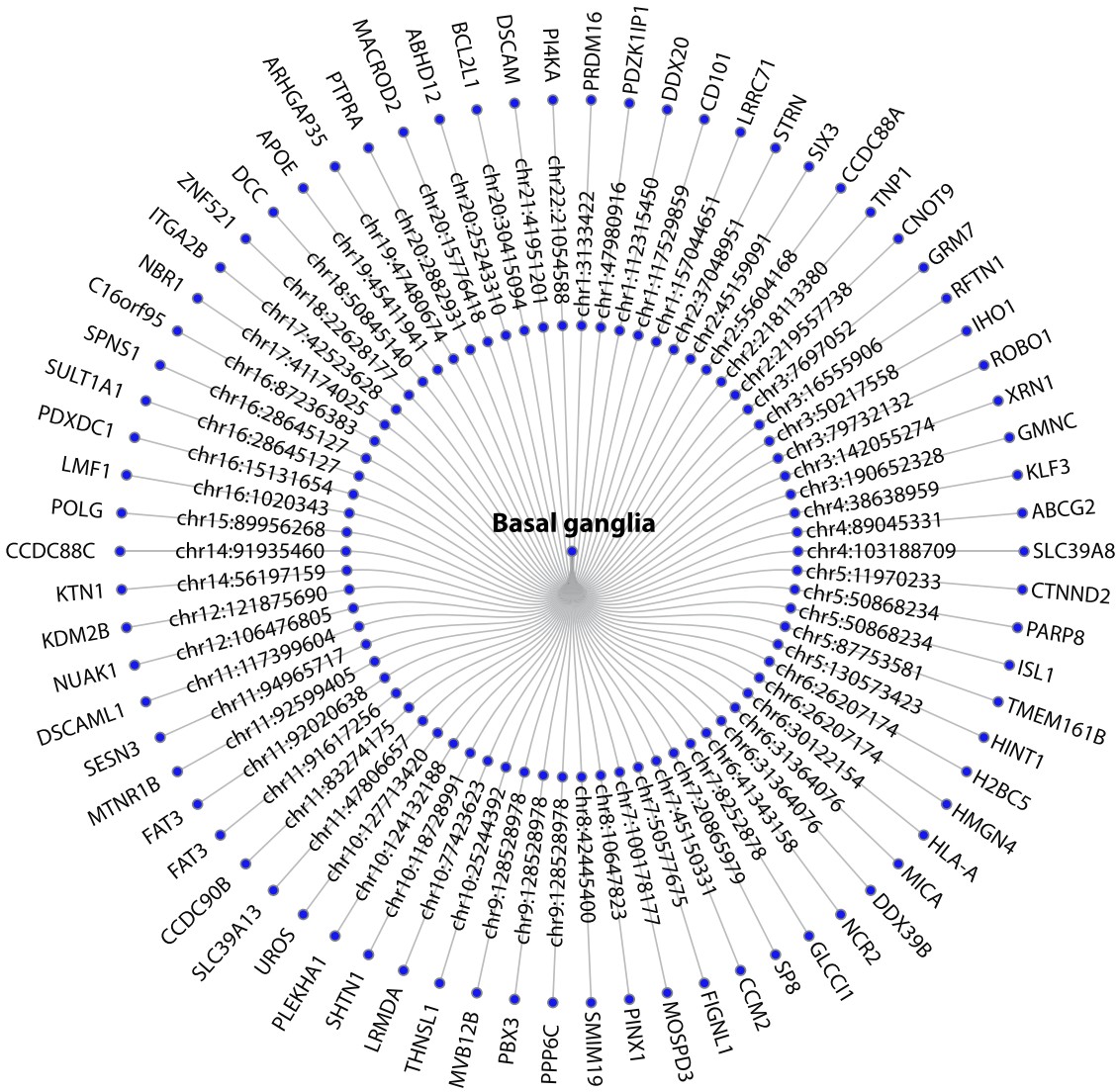

**Fig. 3 | Mapped genes to the significant loci.** Gene mapping of the 72 loci associated with the basal ganglia implied 73 genes by Open target.

the number of significantly shared loci and genes across each combination of disorders. First, we identified significant shared loci according to the FUMA protocol[54] for each conjFDR analyses. Across all phenotypes, we grouped physically overlapping loci, resulting in a total number of 159 distinct loci including 103 loci that were associated with psychiatric disorders and 73 loci associated with neurological disorders. In total, 17 loci were shared between neurological and psychiatric disorders.

Next, we mapped each of these shared loci to significantly associated protein-coding genes using Open target mapping (Supplementary Data 11–18). Overall, the extent of gene pleiotropy was similar to that observed for loci. A total of 165 distinct genes were identified across all phenotypes. Of these, 106 and 75 genes were associated with psychiatric and neurological disorders, respectively, of which 16 were overlapping (Table 1).

We also examined genes that were implicated in multiple disorders. By far the strongest overlap was found between SCZ and MDD, where 8 of the genes overlapping between basal ganglia and SCZ were also found to overlap between basal ganglia and MDD (Fig. 5A). We also found large overlap between other combinations of disorders, such as SCZ and PD (7 genes), and SCZ and BIP (6 genes) (Fig. 5A).

Some of the genes were implicated in more than two disorders. Figure 5B shows all genes overlapping between the basal ganglia

volumes and the 8 disorders. The most frequently mapped genes were the *HP* and *TMEM161B* genes, which overlapped between basal ganglia and at least four disorders. (Fig. 5B).

## Causal interference between the basal ganglia nuclei and common brain disorders

To investigate the potential causal links between basal ganglia nuclei and eight common brain diseases, Mendelian randomization was performed using the same GWAS summary statistics as for conjFDR: ADHD, ASD, BIP, MDD, SCZ, ALZ, MIG, and PD. Individual Mendelian randomization analyses of the disorders with each basal ganglia nuclei (Supplementary Data 19 and 20) identified that the greater genetic liability to ALZ may cause lower accumbens volume (weighted-median: beta = −5.36, $p_{adj}$ = 7.31 × 10⁻³; weighted-mode: beta = −6.14, $p_{adj}$ = 3.40 × 10⁻³; MR-PRESSO: beta = −4.81, $p_{adj}$ = 4.02 × 10⁻²). Meanwhile, there was weak evidence that greater volume of the caudate (IVW: beta = 0.0011, $p_{adj}$ = 6.74 × 10⁻³) and putamen (Egger: beta = 0.0012, $p_{adj}$ = 4.67 × 10⁻²) may increase risk of PD. As both ALZ and PD exhibit a long preclinical disease trajectory, sensitivity analyses were performed using two alternative age cut-offs: 75 and 65 years old, encompassing n = 32,455 and n = 17,591 individuals, respectively. The results remained robust when using the 75-year age cut-off. However, for the 65-year age cut-off, only the association between greater volume of

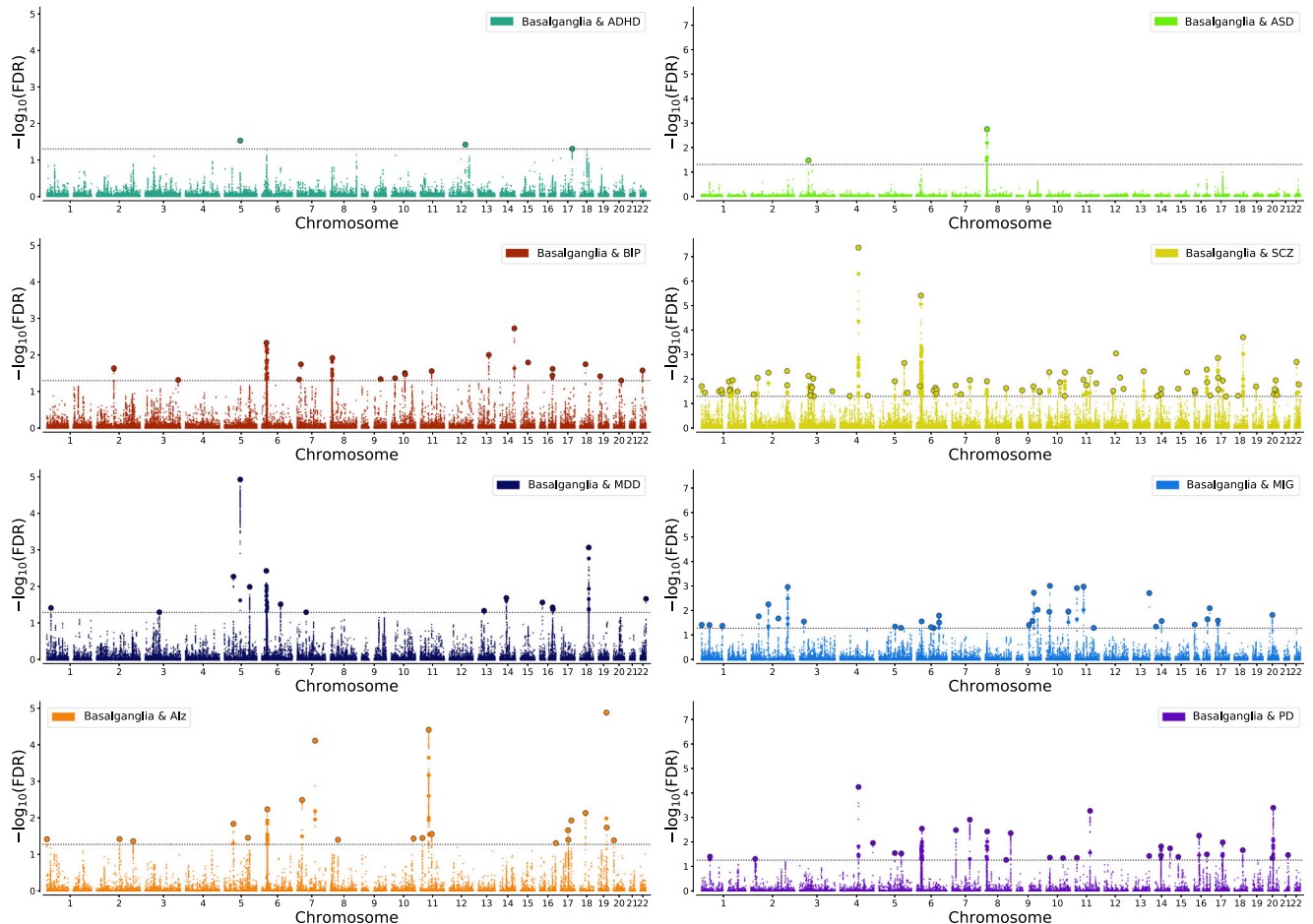

**Fig. 4 | Genetic overlap between basal ganglia and common brain disorders.** Conjunctional FDR Manhattan plots, showing the −log10 transformed conjunctional FDR values for each SNP on the *y*-axis and chromosomal positions along the *x*-axis. The dotted horizontal line represents the threshold for significant shared associations (conjFDR < 0.05). Independent lead SNPs are encircled in black. ASD autism spectrum disorder, ADHD attention deficit hyperactivity disorder, SCZ schizophrenia, BIP bipolar disorder, MIG migraine, MDD, major depression, PD Parkinson's disease, ALZ Alzheimer's disease.

caudate (IVW: beta=0.0011, $p_{adj} = 1.43 \times 10^{-2}$; weighted-median: beta = 0.0014, $p_{adj} = 1.06 \times 10^{-2}$) and the risk of PD remained significant after adjustment for multiple testing (Supplementary Data 21).

## Discussion

In summary, our multivariate GWAS of basal ganglia volumes uncovered 72 genomic loci, 50 of which were not previously identified. Overall, the results indicate that the genetic effects on basal ganglia structures are distributed across basal ganglia structures in line with their function as one unit. The mapped genes have pivotal roles in neurotransmission, neuronal differentiation and synaptogenesis, underscoring their significance in shaping neural circuitry and facilitating brain development. The overlapping genetic architecture and shared genes between basal ganglia and common neurological and psychiatric disorders suggest potential disease-independent drug targets.

Our findings align with and expand upon earlier reports from univariate analyses of studies including basal ganglia volumes together with other subcortical structures. Previous univariate GWASs including basal ganglia nuclei reported five[50] and 29[51] significant loci, respectively, associated with basal ganglia volumes. We replicated 21 loci in our multivariate basal ganglia GWAS and 23 significant loci in our univariate GWASs of accumbens area, caudate, pallidum and putamen. With a total sample size of 30,717, spanning 48 cohorts, Hibar et al.[50] reported reported one locus associated with caudate, four loci with putamen, and none with accumbens or globus pallidus. Satizabal

et al.[51], analyzed a sample of 32,562–37,571, depending on the structure (53 cohorts), and reported four loci associated with accumbens, 10 loci with caudate, six loci with globus pallidus and nine loci with putamen. In the present study, we identified 11 loci associated with accumbens, 26 loci with caudate, 13 loci with globus pallidus and 23 loci with putamen in univariate GWASs on a sample size of 34,794 from the UK Biobank. Of the 95 loci from the five different univariate GWASs, 60 represent unique loci due to a large overlap of loci between the nuclei within the basal ganglia, supporting the interconnectivity between these nuclei. Thus, by leveraging the overlap and integration of function, we were able to increase gene discovery even further with a multivariate GWAS applying the MOSTest. Both our univariate and multivariate GWAS results, represent a large increase in gene discovery from previous work. Our improved results are, beyond the multivariate GWAS approach, likely due to less methodological biases since our study sample is from one cohort, all genotyped with same procedure[66], harmonized imaging data from only four MRI scanners (identical 3 Tesla scanner models, type of coils and protocols), and the same postprocessing protocol across all individuals[67].

Our two most significant loci associated with the basal ganglia have been consistently reported in the previous univariate GWASs on subcortical volumes. One of these loci is located at the 11q14-3 region near the *FAT3* gene, which plays a crucial role in neuronal morphogenesis and cell migration as a conserved cellular adhesion molecule[68]. The second locus is situated at the 14q22 region near the *KTN1* gene, which encodes a kinesin-binding protein involved in the transport of

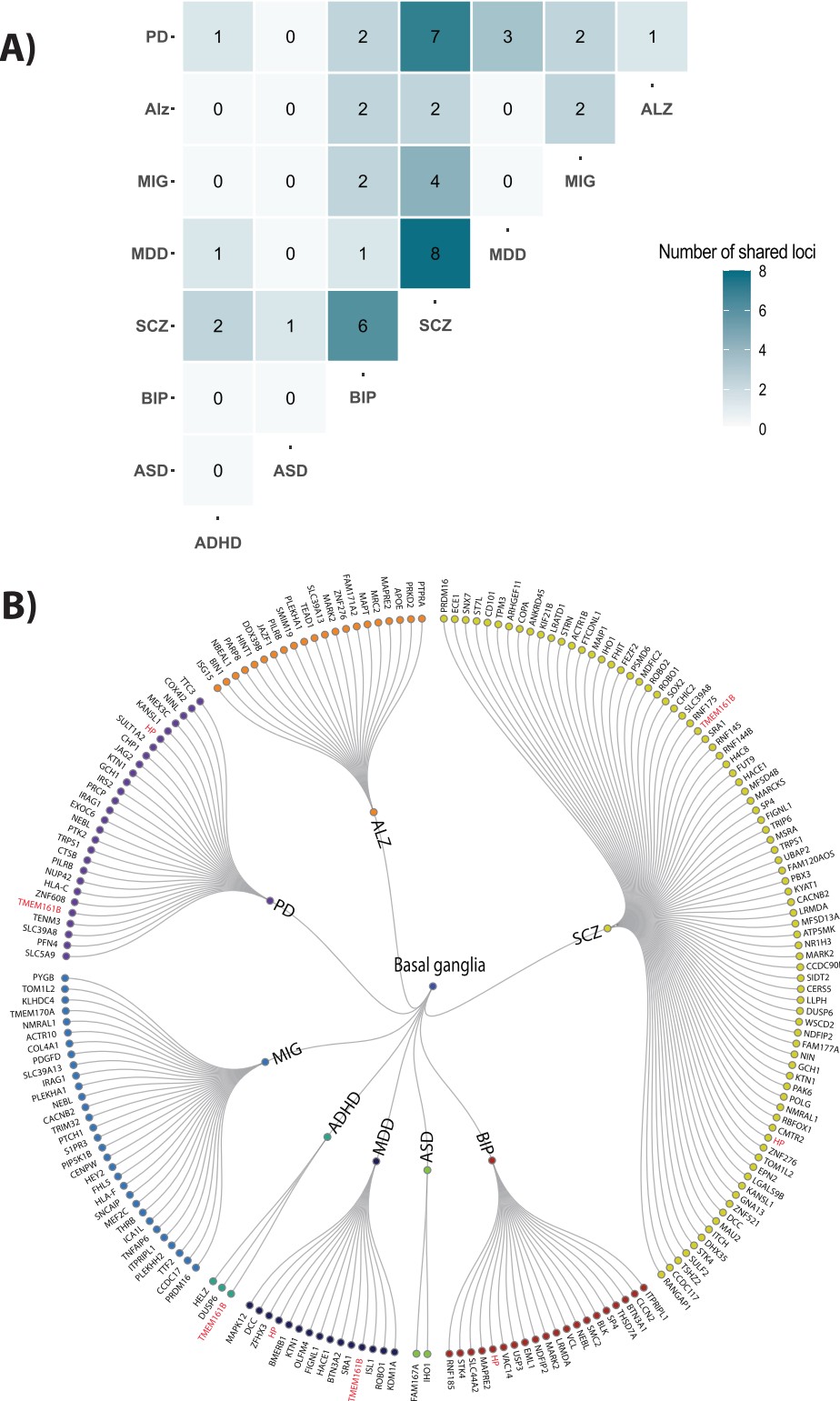

**Fig. 5 | Various genes mapped from the conjunctional FDR analysis were implied to overlap between basal ganglia and multiple disorders. A** The figure shows the total number of genes overlapping for each combination of disorders. For example, 8 of the genes overlapping between basal ganglia and SCZ were also found to overlap between basal ganglia and MDD. **B** All mapped genes to overlapped loci between basal ganglia and disorders and genes that were implied for more than 4 disorders (shown in red). Genes TMEM161B and HP were mapped for four disorders. ASD autism spectrum disorder, ADHD attention deficit hyperactivity disorder, SCZ schizophrenia, BIP bipolar disorder, MIG migraine, MDD, major depression, PD Parkinson's disease, ALZ Alzheimer's disease.

cellular components along microtubules[69]. Earlier univariate GWASs have previously reported associations between *FAT3* and both caudate nucleus[50,51] and putamen[51] volumes, and between *KTN1* and putamen[50,51], in addition to accumbens, caudate nucleus and the globus pallidus[51]. The wide range of associations with basal ganglia volumes contributes to the unquestionable significance of these two loci in our multivariate GWAS analysis. Additional findings that provide support for previous univariate GWAS results include 208 candidate

**Table 1 | Overlapping genes between basal ganglia, psychiatric and neurological disorders**

| Phenotypes | Symbol | CHR | Phenotypes | Symbol | CHR |
|---|---|---|---|---|---|
| ALZ,BIP,SCZ | MARK2 | 11 | MIG,SCZ | TOM1L2 | 17 |
| ALZ, SCZ | ZNF276 | 16 | PD,SCZ | SLC39A8 | 4 |
| ALZ,BIP | MAPRE2 | 18 | ADHD,MDD,PD,SCZ | TMEM161B | 5 |
| MIG,SCZ | PRDM16 | 1 | PD,SCZ | TRPS1 | 8 |
| BIP,MIG | ITPRIPL1 | 2 | PD,SCZ | GCH1 | 14 |
| MIG,SCZ | CACNB2 | 10 | MDD,PD,SCZ | KTN1 | 14 |
| BIP,MIG,PD | NEBL | 10 | BIP,MDD,PD,SCZ | HP | 16 |
| MIG,SCZ | NMRAL1 | 16 | PD,SCZ | KANSL1 | 17 |

*ALZ* Alzheimer's disease, *BIP* bipolar disorder, *SCZ* schizophrenia, *MIG* migraine, *PD* Parkinson's disease, *MDD* major depressive disorder, *ADHD* attention-deficit/hyperactivity disorder.

SNPs annotated to the 8p11.21 region associated with the present basal ganglia GWAS (see Supplementary Data 4), and previously seen associated with putamen in univariate GWAS[51]. Among these, 73 SNPs are mapped to *SLC20A2*, previously linked to the familiar basal ganglia calcification[70–72], although the lead SNP is mapped to *SMIM19* gene, located in close proximity to *SLC20A2*.

The two most significant SNPs, and the above-mentioned SNPs mapped to SLC20A2 and SMIM19, are all located in intergenic regions of the genome. However, four of the 72 basal ganglia associated loci have lead SNPs in the exonic regions (see Supplementary Fig. 4 and Supplementary Data 3): *SLC39A8* on chr 4, *TRIM10* and *HLA-A* on chr 6, *VAT1* and *NBR1* on chr 17 and *APOE* on chr 19. *SLC39A8* has also previously been linked to basal ganglia[51], is a transmembrane protein known to cotransport divalent cations with bicarbonate[73], of which transport of manganese is suggested to be of greatest importance[74], especially for dopaminergic projections in the basal ganglia[75]. The remaining three, however, are novel for this basal ganglia GWAS. rs3094134 on chr 6 located in the exon on *TRIM10* and mapped to *HLA-A*, both important for the immune functioning in the brain[76–78]. The ALZ-linked *APOE* gene on chr 19, known to promote amyloid degradation[79], was associated with basal ganglia volumes, which is a novel finding. Several significant SNPs are annotated to the same region on chr 17 where rs8482 is in the exon of *NBR1*, encoding an autophagic adapter protein involved in Lewy body formation in PD[80] and targeting ubiquitinated protein for degradation in general[81]. The lead SNP in this locus, rs2070835, is located in the exonic region of *VAT1* (vesicle amine transport protein 1), which encodes a protein that plays a crucial role in the process of vesicular neurotransmitter transport[82]. Vat1 is primarily involved in the packaging and storage of monoamine neurotransmitters, such as dopamine, norepinephrine, and serotonin, all neurotransmitters distributed in the basal ganglia. *VAT1* and *NBR1* are located in close proximity to each other on 17q21, on each side of the tumor suppressor gene *BRCA1*. Mapped with Open Target *NBR1* is ranked first and *VAT1* second.

The current boost in discovery with the multivariate MOSTest method supports shared genetic underpinnings[53,83] underlying the interconnected nature of the basal ganglia nuclei. These structures don't operate in isolation, and the finely tuned internal connections within the basal ganglia that are paramount for its function are probably also related to genetic variation, as suggested by the current findings of polygenic pleiotropy across the basal ganglia volumes. By treating multiple interconnected nuclei within the basal ganglia as a collective unit, we capture a wider range of genetic variation, allowing for the detection of loci missed in traditional univariate analyses. This approach not only enhances sensitivity to identify loci with subtle effects but also aligns with the biological rationale of the basal ganglia functioning as a coordinated network, influencing various aspects of motor control, cognition, and emotion. Thus, it is likely that genetic overlap of basal ganglia with various neurological, psychiatric, and

developmental conditions could indicate potential dysregulation. There is clear pleiotropic enrichment between basal ganglia volume and neurological (PD, AD, migraine), neurodevelopmental (ADHD, ASD) and psychiatric (BIP, MDD, SCZ) disorders, shown by Q-Q plots (Supplementary Fig. 13), with many shared loci (Fig. 4, Supplementary Datas 11–18). Notably, the sole significant correlation revealed through LD-score regression-based genetic correlations was a positive association between genetic risk of PD and genetic architecture of basal ganglia (Supplementary Fig. 12 and Supplementary Data 10). Through Mendelian randomization, two independent methods show that genetically determined larger caudate and putamen volume is associated with a slight increase in the risk of PD, supporting the LD-score regression results. As a response to learning, locally increased brain volumes have been reported, possibly representing synaptogenesis and increased dendritic arborization[84,85]. Gene ontology analysis (Supplementary Data 8) support these mechanisms in the genetic architecture of basal ganglia volume, with gene ontology on cell part morphogenesis, neuron part and synapse, but details on cell structure changes representing the positive correlation between the genetic pattern of basal ganglia and PD need separate studies. Additionally, Mendelian randomization showed potential causal effect of ALZ on smaller nucleus accumbens volume and higher striatal volume on PD. The results indicating that genetic predisposition to ALZ influence nucleus accumbens volume is supported by multiple Mendelian randomization methods, increasing the robustness of the findings. In addition to the hallmark feature of medial temporal lobe atrophy, accumbens atrophy has also been found in ALZ patients, correlating with lower cognitive scores[86]. As a limitation, individuals with ALZ and PD are included in the imaging sample, but the confounding effect is negligible as only five and 17 of the 34,794 individuals included in the current study are diagnosed with ALZ and PD, respectively (Supplementary Fig. 14). While implementing sensitivity analysis (Supplementary Data 21) with stringent age filters, we observed an anticipated decline in statistical power; nevertheless, the persistent association between higher striatal volumes and PD genetic markers underscores the potential linkage, as detailed in the supplementary material. Despite the diminished cohort size, the consistency of our PD-related results across age brackets bolsters their validity. Conversely, the non-significant correlation between accumbens volume and AD risk in participants under 65 suggests that our findings may be influenced by preclinical neuropathology affecting this result, challenging the notion of genetic AD risk as a stand-alone causal factor for reduced accumbens volume.

Our results suggest significant genetic pleiotropy between the basal ganglia and especially PD, SCZ and migraine, but also between the basal ganglia and ALZ, BIP and MDD, and less with ASD and ADHD. Out of more than 150 genes, only 16 genes were mapped to basal ganglia and both neurological and psychiatric disorders, indicating distinct genetic architecture underlying each of the disorder's relationship with the basal ganglia. These 16 genes that show overlapping involvement in both neurological and psychiatric disorders, demonstrate important shared biology between brain disorders (Table 1). Notably, *TMEM161B* (transmembrane protein 161B) and *HP* (haptoglobin) is overlapping between basal ganglia and MDD, PD and SCZ, in addition to ADHD and BIP, respectively. Haptoglobin is involved in binding and transporting hemoglobin, and it plays a role in modulating the immune response and oxidative stress[87]. *TMEM161B* has been mapped to MDD in mouse models and humans[88,89] and has also been coupled to basal ganglia activation during reward processing[90]. Although functionally linked to basal ganglia, *TMEM161B* have previously only been linked to neocortex structurally, through gyrification in neocortical development[91].

In summary, our findings indicate a shared polygenic architecture across the basal ganglia, spanning the putamen, caudate, pallidum, and accumbens, with high correlation between the basal ganglia nuclei

and a robust 72 loci when analyzed together, of which 50 are novel. The genetic overlap observed with different brain disorders highlights the interconnectedness of the basal ganglia in neurological (PD, ALZ, migraine) and psychiatric (SCZ, BIP, MDD) conditions, but less for neurodevelopmental conditions (ASD and ADHD). By recognizing the distributed nature of genetic effects across the basal ganglia, functional brain unit, we can gain valuable insights into the mechanisms underlying brain disorders and pave the way for future advancements in this field. Moreover, our study reveals that some of these genetic findings align with known treatment targets, supporting the value of our approach. This suggests shared underlying mechanisms and potential targets for therapeutic interventions. Further research is necessary to fully understand the complex relationship between the basal ganglia and brain disorders and apply this to develop more effective treatments.

## Methods

### Sample and pre-processing of imaging and genetic data

We employed T1 weighted MRI data to investigate basal ganglia volumes. Our primary analysis was based on a dataset consisting of 34,794 genotyped individuals of white British descent from the UK Biobank (age range: 45-82 years, mean: 64.3 years, s.d.: 7.5 years, 52.3% females)[66]. Within our cohort, only a few have registered major neurological and psychiatric conditions (Supplementary Fig. 14). To ensure the robustness of our findings, we also conducted a replication analysis using an independent dataset from UK Biobank consisting of 4808 individuals of British descent (age range: 45-81, mean: 69.1, s.d.: 7.8 years, 50.1 % females) and a generalization analysis using an independent dataset from UK Biobank consisting of 5220 individuals with non-white ethnicity (age range: 45-81, mean: 62.9, s.d.: 7.4 years, 54.1% females).

We used Freesurfer v5.3 to extract the volumes of the accumbens, caudate, pallidum, and putamen[92]. For genetic analyses, we used the UK Biobank v3 imputed genetic data and followed the standard quality control procedures to remove SNPs with an imputation quality score of 0.5, a minor allele frequency <0.005, missingness in more than 10% of individuals, and failing the Hardy-Weinberg equilibrium tests at a P value < $1 \times 10^{-9}$.

### Univariate and multivariate genome-wide association analysis

We calculated the average volume between the left and right hemispheres for each of the accumbens, caudate, pallidum, and putamen as well as the total basal ganglia volume. To address potential confounding factors, we controlled for age, age squared, sex, scanning site, Euler score (which serves as a proxy for image quality), total intracranial volume (ICV), and the 20 genetic principal components for each region comprising the basal ganglia. Furthermore, we conducted additional analyses, controlling for the total volume defined as the sum of the four basal ganglia volumes. We controlled for the same covariates for both white and non-white ethnicity samples. For each basal ganglia volume, we used standard univariate GWAS procedure and annotated the significant loci using FUMA. For each basal ganglia volume, we applied standard univariate GWAS procedure using a linear model in PLINK v1.9, with a MAF filter of 0.005 and annotated the significant loci using FUMA. The min-P approach takes the smallest p-value of each SNP across multiple univariate GWAS, and corrects this for the effective number of traits studied[93]. SNP-based heritability estimates for the basal ganglia volumes, as well as the genetic correlations between these volumes, were estimated using linkage disequilibrium (LD) score regression[94,95].

We further conducted a multivariate GWAS on pre-residualized basal ganglia volumes for our main analysis using Multivariate Omnibus Statistical Test (MOSTest)[53], which is a powerful statistical tool used for joint genetic analysis of multiple traits in large-scale data. It aims to identify genetic variants associated with complex phenotypes influenced by multiple genetic factors with small effects. By analyzing the traits jointly, MOSTest takes a multivariate approach, considering multiple traits simultaneously and leveraging the genetic overlap across different regions and measures of the brain. This approach enhances the statistical power to detect associations by avoiding the stringent multiple comparison correction required in mass-univariate approaches. Van der Meer et al.[53] provide detailed description of the method, while github.com/precimed/mostest provides information on the software implementation. We also conducted a univariate GWAS of the four basal ganglia volumes and whole basal ganglia (extracted from the univariate stream of MOSTest[53]) for comparison to conventional univariate techniques (GWAS).

### Multivariate replication and generalization analysis

We used a multivariate replication approach developed to test if the multivariate pattern of genetic associations identified in discovery analysis is consistent between discovery, replication and generalization samples. This procedure generates a composite score from mass-univariate z-statistics for each locus identified in the multivariate analysis of the discovery sample, and tests for associations of the composite score with genotype in the replication and generalization samples (for mathematical formulation see Loughnan et al.[96]). We report the percent of loci replicating at $P < 0.05$ and the percent of loci showing the same effect direction.

### Functional annotation, gene-based association, gene-set, tissue, and pathway analysis

We submitted the summary statistic to the FUMA platform v1.4.1[54] in order to identify independent genetic loci. We discovered independent significant SNPs at the statistical significance level $P < 5 \times 10^{-8}$ using the 1000GPhase3 EUR as the reference panel. A portion of the independent significant SNPs in linkage equilibrium with one another at $r^2 < 0.1$ were termed lead SNPs, and all SNPs at $r^2 < 0.6$ with each other were considered independent significant SNPs. If two or more lead SNPs located within one LD block (in 250 kb), we merged them into one genomic risk locus. Combined Annotation Dependent Depletion (CADD) scores, which forecast the deleteriousness of SNPs on protein structure/function[57], RegulomeDB scores, which forecast regulatory functions[55], and chromatin states, which illustrate the transcriptional/regulatory effects of chromatin states at the SNP locus, are all used by FUMA to annotate associated SNPs.

We conducted genome-wide gene-based association and gene-set analyses using MAGMA v.1.08[62] in FUMA. MAGMA performs the gene-based association analysis and assigns p-values to individual genes based on the association between genetic variants within or near the gene and the trait of interest. This analysis is conducted using a multiple regression model, which improves statistical performance compared to single-marker analysis[62]. Furthermore, the gene-set analysis using MAGMA enables the assessment of groups of genes that share common biological functions or pathways, which provides insights into the functional and biological mechanisms underlying complex traits. All variants in the major histocompatibility complex (MHC) region were excluded before running the MAGMA analyses.

Gene mapping is a crucial process for understanding the relationship between genetic variants and traits or diseases. To map the significant SNPs to genes, we used variant to gene (v2g) from Open Targets Genetics[97], which provides a comprehensive and systematic approach to prioritize causal variants and identify likely causal genes associated with various phenotypes and diseases. By integrating positional data on the distance between the variant and each gene's canonical transcription start site, eQTL, pQTL, splicingQTL, epigenomic data, and functional prediction, Open Targets maps lead SNPs to genes.

We also conducted Gene Ontology gene-set analysis based on FUMA's gene ontology classification system[54,98], and pathway analysis using Consensus PathDB[99].

## Cell specificity, protein–protein interaction and gene-drug interaction analyses

To investigate the context-specific expression of mapped genes associated with basal ganglia we used WebCSEA (Web-based Cell-type-Specific Enrichment Analysis of Genes). WebCSEA incorporates a curated collection of 111 scRNA-seq panels from various human tissues and 1,355 tissue-cell types across 11 organ systems[100]. We also determined the cell-specific expression within the human cerebral cortex including neurons, fetal and mature astrocytes, oligodendrocytes, microglia/macrophages, and endothelial cells for each mapped gene separately[101].

In addition, we employed GENEmania[101] for functional genomics analysis, to investigate protein-protein interactions (PPIs) between protein-encoded basal ganglia-linked genes and exported the network to Cytoscape 3.10. We identified potential protein clusters and gain insights into the functional relationships between the proteins of interest. Finally, we performed gene-drug interaction analysis using the Drug Gene Interaction Database to gain insights into the potential associations between mapped genes to basal ganglia and drugs, including known drug targets and potential off-target effects[102] to uncover drug targets that specifically influence the basal ganglia.

## Genetic overlap between the basal ganglia and common brain disorders

We studied the genetic overlap between basal ganglia and ASD, ADHD, SCZ, BIP, MDD, MIG, PD, and ALZ. We used GWAS summary statistics from the Psychiatric Genomics Consortium for ADHD[103], ASD[104], BIP[105], MDD[106], SCZ[107] and ALZ[108], from International headache genetics Consortium for MIG[109], and from the International Parkinson Disease Genomics Consortium[110,111] for PD data. All the studies included in the original GWAS were approved by local ethics authorities.

First, we used linkage disequilibrium (LD) score regression[94,95] to estimate genetic correlations between basal ganglia volumes and each of the brain disorders. Then we created conditional quantile-quantile (Q-Q) plots to evaluate the presence of cross-phenotype polygenic enrichment[65,112], conditioning basal ganglia on each brain disorder and vice versa. The conditional FDR method[65,112,113] builds on an empirical Bayesian statistical framework and leverages polygenic overlap between two traits to improve the discovery of genetic loci associated with the traits of interest. The condFDR method estimate the posterior probability that a SNP has no association with the primary trait, given that the $P$ values for that SNP in both the primary and conditional traits are as small as or smaller than the observed $P$ value. Thus, the method improves the detection of genetic variants associated with the primary trait via reranking the test statistics of the primary phenotype based on the strength of the association with the secondary phenotype. Next, we used the conjunctional FDR (conjFDR) approach at FDR 0.05[65,112,113] to identify common genetic loci between basal ganglia and the included brain disorders. This approach is an extension of the conditional FDR and is defined as the maximum of the 2 conditional FDR statistics for a specific SNP and estimates the posterior probability that a SNP is null for either trait or both, given that the $P$ values for both phenotypes are as small as or smaller than the P values for each trait individually. It takes the highest of the two condFDR statistics for a particular SNP, i.e. for trait A conditional on trait B and trait B conditional on trait A. If the p values for both phenotypes are equal to or less than the p values for each trait separately, this reflects an estimate of the posterior probability that a SNP is null for one or both characteristics. More details are found in the original[112,113] and subsequent publication[65]. The conjFDR analysis does incorporate effect directions, in contrast to genetic correlation analysis, and may thus be used to analyze summary data from multivariate GWAS that lack effect directions. Because complex correlations in regions with intricate LD can bias FDR estimation, two genomic regions—the extended major histocompatibility complex genes region (hg19 location Chr 6: 25119106-33854733) and chromosome 8p23.1 (hg19 location Chr 8: 7242715-12483982) for all phenotypes and MAPT region for PD and APOE region for ALZ and ASD, respectively—were excluded from the FDR-fitting procedures. We used the FUMA protocol to annotate independent genomic loci jointly associated with basal ganglia and each brain disorder. We submitted the results of the conjFDR analysis to FUMA v1.5.1[54]. The lead SNPs and candidate SNPs within each genomic locus were used to identify the genomic loci as significant SNPs with an LD r2 >= 0.6 and conjFDR 0.1 and at least one associated independent significant SNP. ConjFDR <0.05 and $r^2 < 0.6$ were used to determine independent significant SNPs, while lead SNPs were defined if they were in approximate linkage equilibrium with each other ($r^2 < 0.1$). Finally, the significant SNPs were mapped to genes using Open Targets[97] as described above.

## Mendelian randomization

To study potential relationships between basal ganglia nuclei and eight common brain disorders, a set of Mendelian randomization analyses were performed. The Mendelian randomization approaches included weighted mode and weighted median[114], Egger regression (Egger)[115] and the inverse-variance weighted model (IVW)[116], in addition to the outlier-corrected MR-PRESSO approach[117]. For these analyses, only genome-wide significant SNPs ($p < 5 \times 10^{-8}$) for the basal ganglia volumes or disorders were used as instruments. Furthermore, we performed a sensitivity analysis to assess the robustness of our Mendelian randomization findings. This analysis involved young individuals with two alternative age cut-offs: 75 and 65 years old, including $n = 32,455$ and $n = 17,591$ individuals, respectively. We used the TwoSampleMR package in R (0.4.26; github.com/MRCIEU/TwoSampleMR)[118].

## Reporting summary

Further information on research design is available in the Nature Portfolio Reporting Summary linked to this article.

## Data availability

In this study we used brain imaging and genetics data from the UK Biobank [https://www.ukbiobank.ac.uk/], and GWAS summary statistics obtained from the Psychiatric Genomics Consortium [https://www.med.unc.edu/pgc/shared-methods/], 23andMe,inc. [https://www.23andme.com/], International headache genetics Consortium (IHGC) [http://www.headachegenetics.org/content/datasets-and-cohorts], the International Genomics of Alzheimer's Project [https://ctg.cncr.nl/software/summary_statistics], and the International Parkinson Disease Genomics Consortium [https://pdgenetics.org/resources]. The latter included 23andMe data, which was made available through 23andMe under an agreement with 23andMe that protects the privacy of the 23andMe participants [https://research.23andme.com/collaborate/#dataset-access/]. The summary statistics for basal ganglia derived in this study is available in GWAS Catalog [https://www.ebi.ac.uk/gwas/studies/GCST90432154]. FUMA results are available online [https://fuma.ctglab.nl/browse/371].

## Code availability

All code and software needed to generate the results is available as part of public resources, specifically MOSTest (https://github.com/precimed/mostest), FUMA, conjunctional FDR and LD score regression (https://github.com/bulik/ldsc).

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

## Acknowledgements

The authors were funded by Norwegian Health Association (S.B.: 22731, K.N.: 25598), the South-Eastern Norway Regional Health Authority (O.A.A.: 2013-123, 2017-112, 2019-108, L.T.W.: 2014-097, 2015-073, 2016-083), the Research Council of Norway (T.K.: 276082, 323961. O.A.A.: 213837, 223273, 248778, 273291, 262656, 229129, 283798, 311993, 324499. L.T.W.: 204966, 249795, 273345). Stiftelsen Kristian Gerhard Jebsen (O.A.A., L.T.W.), the European Research Council under the European Union's Horizon 2020 Research and Innovation program (L.T.W.: 802998, O.A.A.: 847776), NIH (O.A.A.: 1R01MH129742) and the Department of Neurology at Oslo University Hospital (K.N.). The funding bodies had no role in the analysis or interpretation of the data; the preparation, review or approval of the manuscript; nor in the decision to submit the manuscript for publication. This work was performed on the Tjenester for Sensitive Data (TSD) facilities, owned by the University of Oslo, operated and developed by the TSD service group at the University of Oslo, IT-Department (USIT) and on resources provided by UNINETT Sigma2—the National Infrastructure for High Performance Computing and Data Storage in Norway. The research has been conducted using the UK Biobank Resource (access code 27412) and using summary statistics for various brain disorders that partly included 23andMe data. We would like to thank the research participants and employees of UK Biobank, the 23andMe, the Psychiatric Genomics Consortium, International Headache Genetics Consortium, the International Genomics of Alzheimer's Project and International Parkinson Disease Genomics Consortium for contributing summary statistics for this work.

## Author contributions

S.B., K.N., and O.A.A. conceived the study. S.B., J.R., and Z.R. analyzed the data. S.B, and K.N. interpreted the results and spearheaded the writing. S.B. drafted the online methods. S.B., K.N., J.R, A.S., Z.R., O.S., P.P.J., N.P., P.P., K.O.C., T.E., M.T., S.D., A.D., L.T.W., T.K., and O.A.A. gave conceptual input on the methods and/or results and contributed to and approved the final manuscript.

## Competing interests

Dr. Andreassen has received a speaker's honorarium from Lundbeck, Janssen and a consultant for Cortechs.ai. Other authors report no conflicts.
