## [Peer Review File · Nature Communications]

Unveiling the Genetic Landscape of Basal Ganglia: Implications for Common Brain DisordersREVIEWER COMMENTS

Reviewer #1 (Remarks to the Author):

It was a pleasure to review this manuscript which reported on a multivariate GWAS for basal ganglia volumes: accumbens, caudate, pallidum, putamen, and the total basal ganglia. This revealed 72 independent significant loci, spanning 73 genes. The most significant loci replicated previous univariate GWASs on subcortical volumes. Gene-set analyses revealed terms linked to neurogenesis, neuron differentiation, and development. Association was observed between the genetic architecture of the basal ganglia and Parkinson's Disease.

The manuscript has several commendable strengths. The study appropriately controlled for potential neuroimaging confounds. Furthermore, an independent dataset was used to assess replicability and the comparison of multivariate GWAS with univariate GWAS results demonstrated the enhanced sensitivity of a multivariate approach to detect significant loci. The findings were aptly contextualized through analyses considering cell specificity, protein-protein interactions, gene-drug interactions, and the genetic overlap with a range of brain disorders. Overall, I think this work is of interest to the field; however, addressing the issues below would be important in fully realizing this potential.

Authors acknowledge in the introduction that prior genetic studies have examined volumetrics of basal ganglia structures. It might be helpful to make more explicit the similarity in the phenotypes studied here with prior GWAS analyses, such as those by Hibar (2015) and Satizabal et al. (2019), which examined 4 out of 5 phenotypes explored in this study (all brain regions except for the total basal ganglia volume). The key advance in this research lies in the multivariate approach applied, which improved sensitivity for detecting significant loci.

As previous univariate GWAS analyses were conducted on the same brain regions, it would be informative to ascertain whether any novel loci were detected in the univariate analyses presented here in comparison to prior findings.

The abstract states that a replication rate of 87.5% was obtained, which might be somewhat misleading as only 55.6% of the loci replicated at $P < 0.05$. I recommend clarifying the definition of replication (i.e., results aligning in the same direction) within the abstract or reporting the replication rate based on $P < 0.05$.

The enrichment of immune cell types like natural killer cells and T cells, while somewhat unexpected based on histological analyses, aligns with the growing understanding of the immune system's role in neurological and psychiatric conditions. This observation may be linked to the aging nature of the cohort and the potential influence of medications (highlighted in the gene-drug interaction analysis) or illness-related factors. To aid interpretation, characterizing features in the cohort, including the proportion of subjects diagnosed with major neurological and psychiatric conditions and the portion of subjects taking one of the 130 drugs identified in the gene-drug interaction analysis, would be beneficial.

The abstract mentions that genetic overlap with the basal ganglia was strongest for Parkinson's Disease, schizophrenia, and migraine. While the evidence for Parkinson's Disease is clear and is supported by LD-score regression-based genetic correlations, the position of schizophrenia and migraine in the abstract's ranking isn't as apparent. The mentioned LD-score regression-based genetic correlations did not reveal significant associations with other brain disorders after Bonferroni correction. Instead, the evidence for schizophrenia and migraine was based on the number of loci overlap between basal ganglia loci and risk loci for these disorders (from what I understand). It's worth considering whether this approach could introduce bias due to variations in the number of loci associated with each disorder.

A minor point concerns the replication, which was performed on subjects with non-European ancestry within the UK Biobank dataset. This sample seems more like a 'generalization' sample rather than a 'replication' sample, given that differences between discovery and replication could be due to important ancestry effects, in addition to variation from testing an independent sample. Is a similar (European ancestry) replication dataset available to supplement the current generalization analyses? I also suggest disclosing the characteristics of the current replication sample in the abstract (non-European ancestry). Furthermore, it would be beneficial to discuss the current results in terms of generalizability to a mixed non-European ancestry population.

In Supplementary Figure 8, the spatial resolution is insufficient to discern the findings related to cell type analysis.

Reviewer #2 (Remarks to the Author):

Dear editor,

I reviewed the manuscript "Unveiling the Genetic Landscape of Basal Ganglia: Implications for Common Brain Disorders". The authors aimed to identify genetic markers associated with variation in the volume of basal ganglia structures. To this end they performed a multivariate genome-wide association study (GWAS), combining multiple basal ganglia structure in 1 analysis. They found 72 genetic loci associated with this multivariate trait. The authors assessed overlap between identified genetic loci and known genetic loci for neurological and psychiatric disorders.

Originality and novelty

The work is original in the sense that no multivariate GWAS on the combination of basal ganglia regions has been performed thus far. However, the added value over the existing literature is limited. The authors use data that was previously used to study volume of isolated basal ganglia structures (ref. 50), with a similar sample size. That means the current work is a re-analysis of these data. While this is not an issue per se, given the limited added value to the understanding of the genetic mechanisms shaping basal ganglia volumes, in total the added value of the work is small.

The genetic correlation analysis to assess overlap between basal ganglia substructures which includes the 4 structures studies in the present manuscript were performed previously (ref. 50), but not discussed in the manuscript. In ref. 50 the authors already found "strong phenotypic and genetic overlap among most subcortical structures using LDSC methods, consistent with our finding that many of the loci identified have pleiotropic effects on the volumes of several subcortical structures."

Multivariate versus univariate

The authors started out with univariate analyses of basal ganglia structures but did not present those data fully. They also did not provide Methods for the univariate analyses. They also glanced over the potential clinical and scientific relevance of such analyses. The authors did state that various basal ganglia structures have different relationships with disease (such as substantia nigra with Parkinson's disease). Univariate analyses may help untangle the role of the basal ganglia in this case, but these results were not discussed.

Statistical power may vary per basal ganglia structure based on imaging accuracy and individual variation. Therefore, currently it is not possible to assess whether identified genetic markers are driven predominantly by certain structures.

While there may some added value in a multivariate approach to the genetics of brain structures, without a good description of the univariate genetic causes, differentiation between meaningful results and, for example, power differences between structures are not possible. The knowledge gap that the authors describe ("Comprehensive understanding of the genetic mechanisms shaping basal ganglia

volumes is lacking") is therefore not addressed sufficiently.

The methods currently utilized do not sufficiently study the basal ganglia as a functional unit but are rather a screen of genetic associations with a combinatorial phenotype of basal ganglia volumes.

Other questions regarding the rationale

The authors conducted several gene set enrichment analyses. It is unclear what the clinical relevance is of the drug target analysis with basal ganglia structures. What does this add over drug target analyses on psychiatric and neurological disorders?

The authors established overlap of basal ganglia-related loci, across their association with disorders. The rationale of this method is insufficiently clear. Genetic overlap between psychiatric traits has been studied extensively. What do they author aim to learn from limiting this overlap to basal ganglia-related loci?

Overlap with diseases

The authors performed analyses to assess the overlap with psychiatric and neurological diseases using a number of techniques that are commonly used for such analyses and which are performed well. They do however not go into detail about how the basal ganglia functioning as a unit relates to neurological and psychiatric diseases in the Discussion section.

Claims incompletely supported by evidence

The authors claim that "In summary, our multivariate GWAS of basal ganglia volumes uncovered 51 genomic loci that were not previously identified, indicating that the effects on basal ganglia structures are distributed across basal ganglia structures in line with their function as one unit." Can the authors comment on whether the observed increase in loci discovered can be attributed to an increase in power due to including multiple regions of which the sizes are merely correlated and thus capturing more size variance?

In lines 165-167 the authors claim that the Q-Q plot in Supplementary Figure 1 shows good test statistics. However, no lambda or LD-score regression intercept were presented. Without at least one of these statistics, the abovementioned claim cannot be made.

Possible improvements

Causality between basal ganglia volumes and disease using Mendelian randomization may aid the understanding of the shared genetics between basal ganglia volumes and disease.

Adding behavioral and/or personality traits to the disease-centered analyses may further help understand the role of the basal ganglia in behavior and disease.

Reproducibility

The conditional analysis methods are briefly explained, but to me the rationale and methods remain in part unclear. What do the authors mean by "leverages cross-phenotype enrichment" as mentioned in the Methods?

Based on these methods, the authors state that "Conditional Q-Q plots conditioning the multivariate statistic of basal ganglia on the disorders and vice versa clearly demonstrated a pattern of pleiotropic enrichment in both directions (Supplementary Fig. 11)". However, they do not explain what makes this pleiotropic enrichment so apparent in Supplementary Figure 11.

Clarifications

Please clarify the statement on lines 156-160. Specifically, what do the authors define as a distributed genetic architecture?

Minor comments

The sentence "It is, however, only directly significant association between genetic risk of PD and

genetic architecture of basal ganglia" appears to be incorrect.

Scientific notation is in the form of $1e-5$, where 1×10^{-5} would be preferred.

The authors found 72 statistically significant loci. They state that 51 are novel. While they may indeed not have been related to subcortical volumes before, these loci are not novel in the sense that the multivariate trait has not been investigated before. I recommend choosing a different term or elaborating.

Response to all reviewers

We would like to thank the reviewers for their constructive feedback that clearly helped improving our manuscript. In brief, our major revisions include:

- We have added the results of the univariate GWASs on the individual subcortical structures, to compare with previous univariate GWAS. The results revealed 60 unique loci, 43 of which were novel. The results of the univariate GWASs are added to the abstract, method, results and discussion section of the manuscript
- We have added a replication analysis using an independent dataset from UK Biobank consisting of 4808 individuals of British descent that produced the same effect direction for 87.5% of the lead SNPs
- We have added Mendelian randomization analyses to investigate the potential causal links between basal ganglia volumes and brain diseases. The results of these analyses are added to the abstract, method, results and discussion section of the manuscript.

We marked all changes in the manuscript related to the reviewers' comments with green colour. Responses to individual reviewer comments are provided on the next pages.

Reviewer response letter:

Reviewer #1 (Remarks to the Author):

It was a pleasure to review this manuscript which reported on a multivariate GWAS for basal ganglia volumes: accumbens, caudate, pallidum, putamen, and the total basal ganglia. This revealed 72 independent significant loci, spanning 73 genes. The most significant loci replicated previous univariate GWASs on subcortical volumes. Gene-set analyses revealed terms linked to neurogenesis, neuron differentiation, and development. Association was observed between the genetic architecture of the basal ganglia and Parkinson's Disease.

The manuscript has several commendable strengths. The study appropriately controlled for potential neuroimaging confounds. Furthermore, an independent dataset was used to assess replicability and the comparison of multivariate GWAS with univariate GWAS results demonstrated the enhanced sensitivity of a multivariate approach to detect significant loci. The findings were aptly contextualized through analyses considering cell specificity, protein-protein interactions, gene-drug interactions, and the genetic overlap with a range of brain disorders. Overall, I think this work is of interest to the field; however, addressing the issues below would be important in fully realizing this potential.

Response: We appreciate the reviewer's thorough and insightful evaluation of our manuscript and the valuable feedback. We appreciate the recognition of the strengths present in our manuscript, and we have taken your comments into consideration and made necessary revisions to enhance the clarity and depth of our manuscript.

Authors acknowledge in the introduction that prior genetic studies have examined volumetrics of basal ganglia structures. It might be helpful to make more explicit the similarity in the phenotypes studied here with prior GWAS analyses, such as those by Hibar (2015) and Satizabal et al. (2019), which examined 4 out of 5 phenotypes explored in this study (all brain regions except for the total basal ganglia volume). The key advance in this research lies in the multivariate approach applied, which improved sensitivity for detecting significant loci. As previous univariate GWAS analyses were conducted on the same brain regions, it would be informative to ascertain whether any novel loci were detected in the univariate analyses presented here in comparison to prior findings.

Response: We appreciate the recommendation concerning the clarification of similarities and differences between our study and prior univariate GWAS analyses on subcortical structures encompassing but not limited to the basal ganglia (Hibar et al 2015 and Satizabal et al 2019). Although the key novelty introduced in our research lies in the comprehensive multivariate analysis concentrating specifically on the entirety of the basal ganglia, we agree that we should also report the results of the univariate GWASs on the individual subcortical structures, to compare with previous univariate GWAS. To address this in the manuscript, we have revised the abstract, introduction, results and discussion to explicitly highlight the distinction between the univariate subcortical structure analyses performed in prior studies and the univariate GWASs performed in our study. As a main figure, we have kept it as part of the

Miami plot, while the individual Manhattan plots and detailed results are added to the supplementary (Supplementary Fig. 1 & Supplementary Table 1). Furthermore, we have dedicated a specific section to discuss the outcomes of the univariate GWAS analyses conducted in our study.

For clarity, the main changes to the manuscript is added below:

Introduction section from line 136-140, page 4-5:

“Hibar and colleagues¹ reported one locus significantly associated with caudate nucleus and four loci significantly associated with putamen, but none with accumbens area or globus pallidus. Satizabal and colleagues², on the other hand, reported four loci associated with the accumbens area, 10 loci with the caudate nucleus, six loci with globus pallidus and nine loci with putamen”

Results section from line 153, page 5-6:

“Univariate GWASs reveal 47 novel genetic loci associated with basal ganglia

Univariate GWAS analyses on individual nuclei collectively known as the basal ganglia revealed 60 unique loci, 43 of which were novel. We identified 11 loci associated with accumbens area (Supplementary Table 1, Supplementary figure 1), where three out of the four loci reported by Satizabal et al² are replicated, leaving eight novel loci. We report 26 genetic loci significantly associated with the caudate nucleus (Supplementary table 1, Supplementary figure 1), of which 18 loci are novel, due to replication of eight out of the ten loci reported by Satizabal et al², while the one locus reported by Hibar et al¹ is not replicated. For pallidum, we report 13 significantly associated genetic loci (Supplementary table 1, Supplementary figure 1), nine of which are novel, while four are replications of the six loci reported by Satizabal et al². For putamen, we replicate all four loci reported associated with putamen by Hibar et al¹ and seven out of the nine loci reported by Satizabal et al², in addition to 15 novel loci, altogether 23 loci, significantly associated with putamen in our univariate GWAS (Supplementary table 1, Supplementary Fig. 1). We have also performed a univariate GWAS on basal ganglia as a functional unit (one volume), resulting in 22 significant loci spanning 12 chromosomes (Supplementary table 1, Supplementary figure 1). Of these loci, 14 have not been previously reported from the univariate GWASs from the individual nuclei of which basal ganglia consists^{1,2}. There is a large overlap between the loci identified in the univariate GWASs of the different basal ganglia nuclei (mean 22%, range 9-38%) and between the univariate GWAS of the basal ganglia as one volume and univariate GWASs of the different basal ganglia nuclei (mean 42%, range 27-65%), supporting the basal ganglia as a functional group of very intertwined subcortical nuclei.”

Discussion section from line 339 page 13 :

“Previous univariate GWASs including basal ganglia nuclei reported five¹ and 29² significant loci, respectively, associated with basal ganglia volumes. We replicated 21 loci in our multivariate basal ganglia GWAS and 23 significant loci in our univariate GWASs of accumbens area, caudate, pallidum and putamen. With a total sample size of 30,717, spanning 48 cohorts, Hibar et al¹ reported reported one locus associated with caudate, four loci with putamen, and none with accumbens or globus pallidus. Satizabal et al², analyzed a sample of 32,562 – 37,571, depending on the structure (53 cohorts), and reported four loci associated with accumbens, 10 loci with caudate, six loci with globus pallidus and nine loci with putamen. In the present

study, we identified 11 loci associated with accumbens, 26 loci with caudate, 13 loci with globus pallidus and 23 loci with putamen in univariate GWASs on a sample size of 34,794 from the UK Biobank. Of the 95 loci from the five different univariate GWASs, 60 represent unique loci due to a large overlap of loci between the nuclei within the basal ganglia, supporting the interconnectivity between these nuclei. Thus, by leveraging the overlap and integration of function, we were able to increase gene discovery even further with a multivariate GWAS applying the MOSTest. Both our univariate and multivariate GWAS results, represent a large increase in gene discovery from previous work. Our improved results are, beyond the multivariate GWAS approach, likely due to less methodological biases since our study sample is from one cohort, all genotyped with same procedure³, harmonized imaging data from only four MRI scanners (identical 3 Tesla scanner models, type of coils and protocols), and the same postprocessing protocol across all individuals⁴.”

The abstract states that a replication rate of 87.5% was obtained, which might be somewhat misleading as only 55.6% of the loci replicated at $P < 0.05$. I recommend clarifying the definition of replication (i.e., results aligning in the same direction) within the abstract or reporting the replication rate based on $P < 0.05$.

Response: We thank the reviewer for this comment, and we have now revised the abstract to provide a clearer and more accurate representation of the replication outcomes, emphasizing both results that concur in the same direction. Furthermore, we have revised the replication rate to accurately reflect the percentage of loci that replicated at $P < 0.05$, ensuring a more precise and transparent presentation of our findings.

Expanded explanation in abstract from line 55, page 2:

“Our multivariate GWAS identified 72 genetic loci associated with basal ganglia volumes with a replication rate of 55.6% at $P < 0.05$ and 87.5% showed the same direction, revealing a distributed genetic architecture across basal ganglia structures.”

The enrichment of immune cell types like natural killer cells and T cells, while somewhat unexpected based on histological analyses, aligns with the growing understanding of the immune system's role in neurological and psychiatric conditions. This observation may be linked to the aging nature of the cohort and the potential influence of medications (highlighted in the gene-drug interaction analysis) or illness-related factors. To aid interpretation, characterizing features in the cohort, including the proportion of subjects diagnosed with major neurological and psychiatric conditions and the portion of subjects taking one of the 130 drugs identified in the gene-drug interaction analysis, would be beneficial.

Response: We appreciate the reviewer's insightful comment. To provide a more comprehensive understanding, we have made significant additions to the revised manuscript. In the Methods section, we now explicitly state that within our cohort, only a few individuals (eg. five with AD and 17 with PD) have registered major neurological and psychiatric conditions (Supplementary Fig. 14, a detailed figure in the supplementary information illustrating the relevant diagnoses). Additionally, in the Results section, we've emphasized the availability of data on drug use for 60 out of the 124 drugs included in the gene-drug

interaction analysis within the UK Biobank. The number of individuals in our cohort using any of these 60 drugs is now depicted in Supplementary Fig. 11 for better characterization (please refer to Supplementary Fig. 11 for details). We believe that these additions significantly enhance the interpretability of our findings and provide a more nuanced view of the cohort's characteristics.

Result section line 265-267, page 10:

“Within the UK Biobank, data on drug use are available for 60 out of the 124 drugs included in the gene-drug interaction analysis. The number of individuals in our cohort using any of these 60 drugs are illustrated for relevant characterization of our cohort (Supplementary Fig. 11).”

Supplementary Fig. 11. Amount of individuals using any of the 60 drugs both being part of the drug-gene analysis and for which drug use information is available within the UK Biobank.

In our cohort of 34,794 individuals, 4963 are registered as currently or previous users of aspirin, 4351 with simvastatin, 3941 with atorvastatin, 529 using allopurinol, 452 sertraline, 347 using pravastatin, 332 using warfarin, 320 using rosuvastatin, 276 using irbesartan, 251 using methotrexate, 174 using tamoxifen, 168 budesonide, 127 using terbinafine, 67 using sulfasalazine, 56 using fenofibrate, 53 using lamotrigine, 46 using amoxicillin, 42 using carbamazepine, 38 using prednisolone, 34 using nitrofurantoin, 23 using Fluvastatin, 21 using nicotine, 20 using levetiracetam, 17 using triamcinolone, 17 using nedocromil, 16 using tenofovir, 9 using ritonavir, ketoconazole and intraconazole, 8 using nevirapine, minoxidil and efavirenz, 7 using chlorpromazine,

6 using lorazepam, 5 using tacrolimus, risperidone, imatinib and dabsone, 4 using prazosin, lamivudine and fluorouracil, 2 using leucovorin, 1 using zidovudine, virconazole, sirolimus, ribavirin, ganciclovir, cyclophosphamide, apicillin, acetaminophen and acenocoumarol and finally no registred users of vancomycin, troglitazone, ticlopidine, flunisolide, enoxaparin, clavulanic acid, seftriaxone, capecitabine and bacampicillin

Discussion line 450-453, page 16:

“As a limitation, individuals with ALZ and PD are included in the imaging sample, but the confounding effect is negligible as only five and 17 of the 34,794 individuals included in the current study are diagnosed with ALZ and PD, respectively (Supplementary figure 14).”

Supplementary Fig. 14. Within our cohort of 34,794 individuals, only a few have registered major neurological and psychiatric conditions. To elaborate, 1477 individuals have recorded mood disorders, with 1423 having experienced major depression and 82 diagnosed with bipolar disorder. Primary headache disorder is noted in 603 individuals, with 535 of them specifically diagnosed with migraine. Schizophrenia spectrum disorder was identified in 50 patients, ADHD in one individual, ASD in 17 individuals, epilepsy in 267, MS in 110, PD in 876, and AD in 52. Diagnoses according to the ICD-10 framework are registered for all individuals previously being, but outpatient treatments lack such documentation.

The abstract mentions that genetic overlap with the basal ganglia was strongest for Parkinson’s Disease, schizophrenia, and migraine. While the evidence for Parkinson’s Disease is clear and is supported by LD-score regression-based genetic correlations, the position of schizophrenia and migraine in the abstract’s ranking isn’t as apparent. The mentioned LD-score regression-based genetic correlations did not reveal significant

associations with other brain disorders after Bonferroni correction. Instead, the evidence for schizophrenia and migraine was based on the number of loci overlap between basal ganglia loci and risk loci for these disorders (from what I understand). It's worth considering whether this approach could introduce bias due to variations in the number of loci associated with each disorder.

Response: We appreciate the astute observation regarding the abstract's emphasis on genetic overlap with Parkinson's disease, schizophrenia, and migraine. In response to this, we have re-evaluated and refined the abstract to reflect the strongest genetic overlap specifically with Parkinson's Disease, as supported by robust LD-score regression-based genetic correlations.

Abstract line 60 and 61, page 2:

"The strongest genetic overlap was between basal ganglia and Parkinson's disease, as supported by robust LD-score regression-based genetic correlations."

A minor point concerns the replication, which was performed on subjects with non-European ancestry within the UK Biobank dataset. This sample seems more like a 'generalization' sample rather than a 'replication' sample, given that differences between discovery and replication could be due to important ancestry effects, in addition to variation from testing an independent sample. Is a similar (European ancestry) replication dataset available to supplement the current generalization analyses? I also suggest disclosing the characteristics of the current replication sample in the abstract (non-European ancestry). Furthermore, it would be beneficial to discuss the current results in terms of generalizability to a mixed non-European ancestry population.

Response: We appreciate the reviewer's insightful comments and have taken them into careful consideration. In response to the concern raised, we conducted an additional multivariate GWAS analysis for replication using a cohort of white European ancestry (n=4,808). This new analysis supplements our previous generalization analysis on subjects with non-white European ancestry within the UK Biobank dataset (n=5,220). The replication dataset now consists of individuals of white European ancestry. To address the reviewer's suggestion, we have included information about the characteristics of the generalization and replication samples in the methods, in addition to abstract and results. We believe that these enhancements significantly strengthen the rigor and applicability of our study, and we are grateful for the valuable input provided by the reviewer.

Abstract line 52-57:

"We conducted univariate and multivariate genome-wide association analyses (GWAS) to explore the genetic architecture of basal ganglia volumes using brain scans obtained from 34,794 white Europeans with replication in 4,808 white and generalization in 5,220 non-white Europeans. Our multivariate GWAS identified 72 genetic loci associated with basal ganglia volumes with a replication rate of 55.6% at $P < 0.05$ and 87.5% showed the same direction, revealing a distributed genetic architecture across basal ganglia structures."

Results line 210-212:

“The findings were supported by a multivariate replication study from white UK Biobank and a multivariate generalization (non-white) study that produced the same effect direction for 87.5% of the lead SNPs (Supplementary Fig. 3).”

Methods line 901-906, page 28:

“To ensure the robustness of our findings, we also conducted a replication analysis using an independent dataset from UK Biobank consisting of 4,808 individuals of British descent (age range: 45-81, mean: 69.1, s.d.: 7.8 years, 50.1 % females) and a generalization analysis using an independent dataset from UK Biobank consisting of 5,220 individuals with non-white ethnicity (age range: 45–81, mean: 62.9, s.d.: 7.4 years, 54.1% females). ”

Additionally, we have now added a new supplementary figure 3, illustrating replication and generalization.

Supplementary Fig. 3. Two Multivariate replication analyses using independent data from 4,808 individuals of white European ancestry and 5,220 individuals of non-white European ancestry. Using a multivariate replication procedure (see Methods), in both cohorts we found that 55.6% of the loci replicated at $P < 0.05$ and 87.5% showed the same direction. P-values are denoted as $-\log_{10}(P)$.

In Supplementary Figure 8, the spatial resolution is insufficient to discern the findings related to cell type analysis.

Response: We appreciate the reviewer's observation regarding the spatial resolution in Supplementary Figure 8. In response to this feedback, we have made substantial improvements by adjusting the figure layout. We have increased the size of the figure from one page to three pages and reduced the number of columns from five to three to enhance the spatial resolution. We believe these modifications significantly improve the clarity and discernibility of the cell type analysis findings.

Reviewer #2 (Remarks to the Author):

I reviewed the manuscript “Unveiling the Genetic Landscape of Basal Ganglia: Implications for Common Brain Disorders”. The authors aimed to identify genetic markers associated with variation in the volume of basal ganglia structures. To this end they performed a multivariate genome-wide association study (GWAS), combining multiple basal ganglia structure in 1 analysis. They found 72 genetic loci associated with this multivariate trait. The authors assessed overlap between identified genetic loci and known genetic loci for neurological and psychiatric disorders.

Originality and novelty

The work is original in the sense that no multivariate GWAS on the combination of basal ganglia regions has been performed thus far. However, the added value over the existing literature is limited. The authors use data that was previously used to study volume of isolated basal ganglia structures (ref. 50), with a similar sample size. That means the current work is a re-analysis of these data. While this is not an issue per se, given the limited added value to the understanding of the genetic mechanisms shaping basal ganglia volumes, in total the added value of the work is small.

The genetic correlation analysis to assess overlap between basal ganglia substructures which includes the 4 structures studies in the present manuscript were performed previously (ref. 50), but not discussed in the manuscript. In ref. 50 the authors already found “strong phenotypic and genetic overlap among most subcortical structures using LDSC methods, consistent with our finding that many of the loci identified have pleiotropic effects on the volumes of several subcortical structures.”

Multivariate versus univariate

The authors started out with univariate analyses of basal ganglia structures but did not present those data fully. They also did not provide Methods for the univariate analyses. They also glanced over the potential clinical and scientific relevance of such analyses. The authors did state that various basal ganglia structures have different relationships with disease (such as substantia nigra with Parkinson’s disease). Univariate analyses may help untangle the role of the basal ganglia in this case, but these results were not discussed.

Response: We thank the reviewer for taking the time to review our manuscript. We value the recognition of the novelty in our approach, focusing on the basal ganglia as a functional unit (one volume), and the acknowledgment of the originality attributed to our study in employing a unique multivariate method.

We also appreciate the reviewer's concerns, but we respectfully disagree that the added value is limited. However, we agree it's important to clarify that although our study includes a similar sample size as Satizabal et al. 2019², we only have a small sample overlap. In the Satizabal et al.'s study approximately 20% of the sample was from the UK Biobank cohort (1 of 53 cohorts). Meanwhile, our work consists entirely of UK Biobank data, a single cohort with less methodological heterogeneity. Also, our segmentation is consistent throughout the cohort, giving less methodological variance. We have added this clarification to the discussion to ensure transparency and clarify the novelty.

To further clarify the added value of our work, we have included the univariate GWAS results in the revised manuscript, allowing for a comparison with previous univariate GWAS results. In summarizing the comparisons between our univariate GWAS results and Satizabal et al. 2019, there is a clear increase in discovery of novel genetic loci:

- Accumbens area: 11 vs. 3 significant loci (3.6 x increase)
- Caudate nucleus: 26 vs. 10 significant loci (2.6 x increase)
- Pallidum: 13 vs. 6 significant loci (2.2 x increase)
- Putamen: 23 vs. 9 significant loci (2.6 x increase)

Additionally, we have conducted the first univariate GWAS on the combined volume of basal ganglia, identifying 22 significant loci spanning 12 chromosomes. Notably, 14 of these loci have never been reported in previous univariate GWAS on the individual nuclei that constitute the basal ganglia (see Supplementary Fig. 1 & Supplementary Table 1).

To address the comment regarding the univariate analyses, we have now provided a comprehensive description of the univariate GWAS in the methods, results, and discussion sections. We believe these additions enhance the overall clarity and value of our work.

Discussion line 339-359, page 13:

“Previous univariate GWASs including basal ganglia nuclei reported five¹ and 29² significant loci, respectively, associated with basal ganglia volumes. We replicated 21 loci in our multivariate basal ganglia GWAS and 23 significant loci in our univariate GWASs of accumbens area, caudate, pallidum and putamen. With a total sample size of 30,717, spanning 48 cohorts, Hibar et al¹ reported one locus associated with caudate, four loci with putamen, and none with accumbens or globus pallidus. Satizabal et al², analyzed a sample of 32,562 – 37,571, depending on the structure (53 cohorts), and reported four loci associated with accumbens, 10 loci with caudate, six loci with globus pallidus and nine loci with putamen. In the present study, we identified 11 loci associated with accumbens, 26 loci with caudate, 13 loci with globus pallidus and 23 loci with putamen in univariate GWASs on a sample size of 34,794 from the UK Biobank. Of the 95 loci from the five different univariate GWASs, 60 represent unique loci due to a large overlap of loci between the nuclei within the basal ganglia, supporting the interconnectivity between these nuclei. Thus, by leveraging the overlap and integration of

function, we were able to increase gene discovery even further with a multivariate GWAS applying the MOSTest. Both our univariate and multivariate GWAS results, represent a large increase in gene discovery from previous work. Our improved results are, beyond the multivariate GWAS approach, likely due to less methodological biases since our study sample is from one cohort, all genotyped with same procedure³, harmonized imaging data from only four MRI scanners (identical 3 Tesla scanner models, type of coils and protocols), and the same postprocessing protocol across all individuals⁴”

Statistical power may vary per basal ganglia structure based on imaging accuracy and individual variation. Therefore, currently it is not possible to assess whether identified genetic markers are driven predominantly by certain structures.

While there may some added value in a multivariate approach to the genetics of brain structures, without a good description of the univariate genetic causes, differentiation between meaningful results and, for example, power differences between structures are not possible. The knowledge gap that the authors describe (“Comprehensive understanding of the genetic mechanisms shaping basal ganglia volumes is lacking”) is therefore not addressed sufficiently.

The methods currently utilized do not sufficiently study the basal ganglia as a functional unit but are rather a screen of genetic associations with a combinatorial phenotype of basal ganglia volumes.

Response: We appreciate the reviewer's consideration of statistical power variations across basal ganglia structures and the importance of understanding the contributions of individual nuclei. While we acknowledge the potential influence of imaging accuracy and individual variation on statistical power, we would like to highlight certain aspects that address these concerns.

1. *Multivariate Approach Clarification:* The multivariate approach adopted in our study allows us to simultaneously examine the genetic influences on the different basal ganglia nuclei. By doing so, we aim to capture shared genetic factors influencing the volumes of these nuclei. This approach is valuable for uncovering overarching genetic patterns contributing to the collective phenotype of basal ganglia volumes.
2. *Inclusion of Univariate Analyses:* In response to the reviewer's feedback, we have now included more detailed univariate GWAS analyses in our study. These analyses offer a comprehensive exploration of the genetic causes for each basal ganglia nucleus separately. The results of these univariate analyses have been incorporated into our manuscript, providing a thorough examination of the genetic influences on individual structures.
3. *Power and Comprehensive Understanding:* While variations in statistical power between structures are inherent, the combination of both multivariate and univariate analyses enhances the robustness of our findings. This approach allows us to identify shared and unique genetic contributors to basal ganglia volumes. We acknowledge the reviewer's concern about the potential limitations of our methods, and we believe the inclusion of both analyses contributes to a more comprehensive understanding of the genetic mechanisms shaping basal ganglia volumes.

To specify, the increase in the number of discovered genomic loci in our multivariate analysis is almost two times higher compared to the Min-P approach analysis. In summary, the adoption of both multivariate and univariate analyses addresses the concern raised by the reviewer, and we have updated our manuscript to reflect these additional analyses and clarified how this approach utilized the basal ganglia as a functional unit.

Clairification of the study aim and inclusion of both univariate and multivariate GWAS in the introduction line 144-146:

“Thus, there are likely pleiotropic genetic variants shared across the different nuclei, which can be identified with the multivariate MOSTest method ⁵. Still, we also include univariate GWASs for comparisons with earlier studies ^{1,2}.”

Inclusion of detailed results for univariate GWAS of individual basal ganglia nuclei on line 153-170, page 5 and 6:

“Univariate GWASs reveal 47 novel genetic loci associated with basal ganglia

Univariate GWAS analyses on individual nuclei collectively known as the basal ganglia revealed 60 unique loci, 43 of which were novel. We identified 11 loci associated with accumbens area (Supplementary Table 1, Supplementary figure 1), where three out of the four loci reported by Satizabal et al ² are replicated, leaving eight novel loci. We report 26 genetic loci significantly associated with the caudate nucleus (Supplementary table 1, Supplementary figure 1), of which 18 loci are novel, due to replication of eight out of the ten loci reported by Satizabal et al ², while the one locus reported by Hibar et al ¹ is not replicated. For pallidum, we report 13 significantly associated genetic loci (Supplementary table 1, Supplementary figure 1), nine of which are novel, while four are replications of the six loci reported by Satizabal et al ². For putamen, we replicate all four loci reported associated with putamen by Hibar et al ¹ and seven out of the nine loci reported by Satizabal et al ², in addition to 15 novel loci, altogether 23 loci, significantly associated with putamen in our univariate GWAS (Supplementary table 1, Supplementary Fig. 1). We have also performed a univariate GWAS on basal ganglia as a functional unit (one volume), resulting in 22 significant loci spanning 12 chromosomes (Supplementary table 1, Supplementary figure 1). Of these loci, 14 have not been previously reported from the univariate GWASs from the individual nuclei of which basal ganglia consists ^{1,2}. There is a large overlap between the loci identified in the univariate GWASs of the different basal ganglia nuclei (mean 22%, range 9-38%) and between the univariate GWAS of the basal ganglia as one volume and univariate GWASs of the different basal ganglia nuclei (mean 42%, range 27-65%), supporting the basal ganglia as a functional group of very intertwined subcortical nuclei.”

Multivariate approach clarification in discussion, eg line 352-353, page 13:

“Thus, by leveraging the overlap and integration of function, we were able to increase gene discovery even further with a multivariate GWAS applying the MOSTest.”

Other questions regarding the rationale

The authors conducted several gene set enrichment analyses. It is unclear what the clinical relevance is of the drug target analysis with basal ganglia structures. What does this add over drug target analyses on psychiatric and neurological disorders?

Response: We appreciate the reviewer's comment regarding the clinical relevance of the drug target analysis specifically applied to basal ganglia structures and how it adds to existing drug target analyses on psychiatric and neurological disorders.

The drug target analysis on basal ganglia structures provides a targeted examination of potential therapeutic avenues related to these specific brain volumes. While drug target analyses on psychiatric and neurological disorders broadly explore a range of brain structures, our focused approach allows for a more detailed understanding of drug targets that specifically influence the basal ganglia.

By honing in on these structures, we aim to uncover potential drug targets that may have more direct implications for disorders or conditions associated with basal ganglia dysfunction. This specialized analysis provides a nuanced perspective and may identify targets that could be overlooked in broader drug target analyses. We have now revised the manuscript to clarify this:

Clairfied in methods in line 996 -1000, page 32:

"Finally, we performed gene-drug interaction analysis using the Drug Gene Interaction Database to gain insights into the potential associations between mapped genes to basal ganglia and drugs, including known drug targets and potential off-target effects⁶ to uncover drug targets that specifically influence the basal ganglia."

Although this analysis is not emphasized as a primary finding in our study, we believe that including it in the supplementary results adds value by providing readers with additional information on potential drug targets related to the basal ganglia.

We have revised the manuscript adding the amount of individuals in our cohorts using the drugs included in the gene-drug analysis.

Results line 265-267, page 10:

"Within the UK Biobank, data on drug use are available for 60 out of the 124 drugs included in the gene-drug interaction analysis. The number of individuals in our cohort using any of these 60 drugs are illustrated for relevant characterization of our cohort (Supplementary Fig. 11)."

The authors established overlap of basal ganglia-related loci, across their association with disorders. The rationale of this method is insufficiently clear. Genetic overlap between psychiatric traits has been studied extensively. What do they author aim to learn from limiting this overlap to basal ganglia-related loci?

Response: We appreciate the reviewer's comment regarding the rationale behind establishing the overlap of basal ganglia-related loci specifically, as opposed to a broader exploration of genetic overlap across psychiatric traits. Our intention in focusing on basal ganglia-related loci was to provide a targeted analysis that could offer insights into shared genetic factors specifically associated with the basal ganglia, as a functional brain unit (Nelson & Kreitzer et al, 2014).

While we agree that genetic overlap among psychiatric traits has been extensively studied, our interest lies in understanding the unique genetic architecture associated with the basal ganglia, and to what extent this genetic architecture is shared with common brain diseases. By narrowing our focus to loci related to the basal ganglia, we aim to uncover potential genetic links that may be distinct or more pronounced within the context of basal ganglia function. This has now been clarified in the revised manuscript.

Discussion from line 415-426, page 15

“The current boost in discovery with the multivariate MOSTest method supports shared genetic underpinnings underlying the interconnected nature of the basal ganglia nuclei. These structures don't operate in isolation, and the finely tuned internal connections within the basal ganglia that are paramount for its function are probably also related to genetic variation, as suggested by the current findings of polygenic pleiotropy across the basal ganglia volumes. By treating multiple interconnected nuclei within the basal ganglia as a collective unit, we capture a wider range of genetic variation, allowing for the detection of loci missed in traditional univariate analyses. This approach not only enhances sensitivity to identify loci with subtle effects but also aligns with the biological rationale of the basal ganglia functioning as a coordinated network, influencing various aspects of motor control, cognition, and emotion. Thus, it is likely that genetic overlap of basal ganglia with various neurological, psychiatric, and developmental conditions could indicate potential dysregulation.”

Overlap with diseases

The authors performed analyses to assess the overlap with psychiatric and neurological diseases using a number of techniques that are commonly used for such analyses and which are performed well. They do however not go into detail about how the basal ganglia functioning as a unit relates to neurological and psychiatric diseases in the Discussion section.

Response: We recognize the importance of adding to the discussion on how the basal ganglia, as a unit, relates to neurological and psychiatric diseases. To address this aspect, we have expanded our Discussion section to highlight the interconnected nature of basal ganglia nuclei and their collective contribution to motor and cognitive functions in the healthy and diseased brain.

Introduction line 83-85, page 3:

“Although the basal ganglia comprise physically distinct entities, the individual nuclei exhibit a robust functional cohesion due to their intricate interconnections, and is often regarded as a functional unit⁷.”

Introduction line 95-97, page 3:

“However, there is strong evidence supporting that the different basal ganglia are integrated, as the function of the individual nuclei depends on input from neighboring nuclei, making basal ganglia an interconnected network⁷. “

Results line 170-174, page 6:

“There is a large overlap between the univariate GWASs of the different basal ganglia nuclei (mean 22%, range 9-38%) and between the univariate GWAS of the basal ganglia as one volume and univariate GWASs of the different basal ganglia nuclei (mean 42%, range 27-65%), supporting the basal ganglia as a functional group of intertwined subcortical nuclei.”

Discussion line 349-353, page 13:

“Of the 95 loci from the five different univariate GWASs, 60 represent unique loci due to a large overlap of loci between the nuclei within the basal ganglia, supporting the interconnectivity between these nuclei. Thus, by leveraging the overlap and integration of function, we were able to increase gene discovery even further with a multivariate GWAS applying the MOSTest.”

Discussion line 415-426, page 15:

“The current boost in discovery with the multivariate MOSTest method supports shared genetic underpinnings underlying the interconnected nature of the basal ganglia nuclei. These structures don't operate in isolation, and the finely tuned internal connections within the basal ganglia that are paramount for its function are probably also related to genetic variation, as suggested by the current findings of polygenic pleiotropy across the basal ganglia volumes. Thus, it is likely that genetic overlap with various neurological, psychiatric, and developmental conditions could indicate potential dysregulation.”

Claims incompletely supported by evidence

The authors claim that “In summary, our multivariate GWAS of basal ganglia volumes uncovered 51 genomic loci that were not previously identified, indicating that the effects on basal ganglia structures are distributed across basal ganglia structures in line with their function as one unit.” Can the authors comment on whether the observed increase in loci discovered can be attributed to an increase in power due to including multiple regions of which the sizes are merely correlated and thus capturing more size variance?

Response: We appreciate the insightful question raised by the reviewer regarding the potential reasons for the observed increase in the number of genomic loci discovered in our multivariate GWAS (MOSTest) analysis of basal ganglia volumes. As also outlined in response to Reviewer 1, we identified 51 new genomic loci associated with basal ganglia volumes, which were not previously reported. This significant increase in the number of loci can be attributed to several factors, including the unique methodology employed by MOSTest and the biological underpinnings of the basal ganglia as a functional unit. Further, as the reviewer points out, the increase in sample size could be a reason. Further, more consistent brain imaging methodology and genotyping could also play a role.

To differentiate between the different potential explanations, we have now included univariate GWAS results of the individual basal ganglia nuclei, and compared with previous results from similar sample sizes. As reported in the revised results section, we find an increase in number of loci (n=47 novel) compared to previous studies using similar sample size (Satizabel et al 2019). However, this is probably due to reduced methodological bias, since our study sample is from one cohort, all genotyped with same procedure, only used four MRI scanners, and the same postprocessing protocol across all individuals. The remaining increase seems to be driven by the *methodological advancements*: MOSTest is a powerful multivariate

GWAS approach designed to analyze multiple brain regions simultaneously. It leverages the correlated nature of these brain regions, which are part of a cohesive functional network—the basal ganglia. By considering these regions as a collective unit, we gain several advantages in terms of statistical power. These include the following:

- *Capturing more variance*: The basal ganglia comprises multiple interconnected nuclei, each with distinct functions. While these nuclei have unique properties, they also share common genetic underpinnings due to their anatomical and functional relationships. By treating them as a single unit, we capture a broader spectrum of genetic variation that contributes to the overall size and function of the basal ganglia. This approach allows us to detect loci that may have been missed in traditional univariate GWAS analyses, which focus on individual regions.
- *Enhanced sensitivity*: Analyzing the basal ganglia as a unit increases the sensitivity to detect loci with subtle effects that might not have been significant in smaller, univariate studies. This enhanced sensitivity is particularly valuable when investigating complex traits influenced by multiple genetic factors.
- *Biological rationale*: The basal ganglia is known to function as a coordinated network, regulating various aspects of motor control, cognition, and emotion. Genetic factors influencing this network may have broad-ranging effects on multiple regions within the basal ganglia. Therefore, it is biologically plausible that genetic variants affecting this functional unit are distributed across its constituent parts.

In conclusion, the observed increase in the number of genomic loci discovered in our study can mainly be attributed to the power of MOSTest, which leverages the correlated nature of basal ganglia regions and captures a broader spectrum of genetic variation across these regions. Furthermore, the biological rationale for treating the basal ganglia as a functional unit supports our approach, as genetic influences on this complex structure are likely to be distributed across its constituent parts. We have now clarified these factors in the revised manuscript, to provide clarity regarding the unique insights gained through our multivariate GWAS analysis and the validity of the additional loci identified. This is also outlined in response to Reviewer 1, first response, and further outlined above.

Univariate GWAS results of the individual basal ganglia nuclei, and compared with previous results from similar sample sizes (results), lines 154-174 , page 5 and 6, as also outlined in response:

“Univariate GWASs reveal 47 novel genetic loci associated with basal ganglia

Univariate GWAS analyses on individual nuclei collectively known as the basal ganglia revealed 60 unique loci, 43 of which were novel. We identified 11 loci associated with accumbens area (Supplementary Table 1, Supplementary figure 1), where three out of the four loci reported by Satizabal et al² are replicated, leaving eight novel loci. We report 26 genetic loci significantly associated with the caudate nucleus (Supplementary table 1, Supplementary figure 1), of which 18 loci are novel, due to replication of eight out of the ten loci reported by Satizabal et al², while the one locus reported by Hibar et al¹ is not replicated. For pallidum, we report 13 significantly associated genetic loci (Supplementary table 1, Supplementary figure 1), nine of which are novel, while four are replications of the six loci reported by Satizabal et al². For putamen, we replicate all four loci reported associated with putamen by Hibar et al¹ and seven out of the nine loci reported by Satizabal et al², in addition to 15 novel loci, altogether 23 loci, significantly associated with putamen in our univariate GWAS

(Supplementary table 1, Supplementary Fig. 1). We have also performed a univariate GWAS on basal ganglia as a functional unit (one volume), resulting in 22 significant loci spanning 12 chromosomes (Supplementary table 1, Supplementary figure 1). Of these loci, 14 have not been previously reported from the univariate GWASs from the individual nuclei of which basal ganglia consists^{1,2}. There is a large overlap between the loci identified in the univariate GWASs of the different basal ganglia nuclei (mean 22%, range 9-38%) and between the univariate GWAS of the basal ganglia as one volume and univariate GWASs of the different basal ganglia nuclei (mean 42%, range 27-65%), supporting the basal ganglia as a functional group of very intertwined subcortical nuclei.”

Reduced methodological bias (discussion line 355-359), page 13:

“Our improved results are, beyond the multivariate GWAS approach, likely due to less methodological biases since our study sample is from one cohort, all genotyped with same procedure³, harmonized imaging data from only four MRI scanners (identical 3 Tesla scanner models, type of coils and protocols), and the same postprocessing protocol across all individuals⁴.”

Discussion line 415-426, page 15, as also outlined in last response:

“The current boost in discovery with the multivariate MOSTest method supports shared genetic underpinnings^{8,9} underlying the interconnected nature of the basal ganglia nuclei. These structures don't operate in isolation, and the finely tuned internal connections within the basal ganglia that are paramount for its function are probably also related to genetic variation, as suggested by the current findings of polygenic pleiotropy across the basal ganglia volumes. By treating multiple interconnected nuclei within the basal ganglia as a collective unit, we capture a wider range of genetic variation, allowing for the detection of loci missed in traditional univariate analyses. This approach not only enhances sensitivity to identify loci with subtle effects but also aligns with the biological rationale of the basal ganglia functioning as a coordinated network, influencing various aspects of motor control, cognition, and emotion. Thus, it is likely that genetic overlap of basal ganglia with various neurological, psychiatric, and developmental conditions could indicate potential dysregulation.”

In lines 165-167 the authors claim that the Q-Q plot in Supplementary Figure 1 shows good test statistics. However, no lambda or LD-score regression intercept were presented. Without at least one of these statistics, the abovementioned claim cannot be made.

Response: We appreciate that the reviewer is pointing out the missing information regarding the Q-Q plot in Supplementary Figure 1 (new Supplementary Figure 2). To address this concern, we have incorporated LD-score regression intercept values (Bulk-Sullivan et al. Nature Genetics 2015) to provide a more robust assessment of genetic inflation. We have employed the score regression intercept as a test of genetic inflation, after the recommendation of Bulk-Sullivan et al. 2105, as it offers a more reliable and accurate assessment of the degree of confounding bias from inflation than LambdaGC.

For the MOSTest summary statistics, the intercept was determined to be 1.024, indicating minimal to no genetic inflation attributable to bias, such as cryptic relatedness and population stratification. This crucial information has now been included in the results section (line 213,

page 7), and we have replaced the Q-Q-plot from the MOSTest analysis with the LD-score regression intercept values (new Supplementary Fig. 2). Additionally, LDSC intercept values for all other univariate GWASs are presented in Supplementary Figure 1, ranging from 0.999 to 1.026, further supporting the absence of significant genetic inflation due to bias in our analyses. These additions strengthen the reliability of our findings and provide a more comprehensive evaluation of the test statistics.

Supplementary Fig. 2. Quantile-quantile (Q-Q) plots from MOSTest analysis. A) The left panel shows signal from MOSTest analysis. B) The right panel (from permutation testing) shows test statistics under null and confirms validity of the MOSTest test statistics.

Possible improvements

Causality between basal ganglia volumes and disease using Mendelian randomization may aid the understanding of the shared genetics between basal ganglia volumes and disease.

Response: We appreciate the reviewer's insight into the potential utility of Mendelian randomization (MR) to elucidate the potential causality between basal ganglia volumes and disease. We agree that MR is a useful approach to explore causal relationships by leveraging genetic variants as instrumental variables.

We have now added Mendelian randomization analyses to investigate the potential causal links between basal ganglia volumes and eight common brain diseases. The results of these analyses are added to the abstract, method, results and discussion section of the manuscript, in addition to the supplementary tables 19 and 20. We agree that incorporating MR adds valuable information to our exploration of the shared genetics between basal ganglia volumes and disease.

Abstract line 61-64, page 2:

“Mendelian randomization indicated genetic liability to larger striatal volume as potentially causal for Parkinson’s disease, in addition to a suggestive causal effect of greater genetic liability to Alzheimer’s disease on smaller accumbens.”

Methods lines 1032-1039, page 34:

“Mendelian randomization

To study potential relationships between basal ganglia nuclei and eight common brain disorders, a set of Mendelian randomization analyses were performed. The Mendelian randomization approaches included weighted mode and weighted median¹⁰, Egger regression (Egger) ¹¹ and the inverse-variance weighted model (IVW) ¹², in addition to the outlier-corrected MR-PRESSO approach ¹³. For these analyses, only genome-wide significant SNPs ($p < 5 \times 10^{-8}$) for the basal ganglia volumes or disorders were used as instruments. We used the TwoSampleMR package in R (0.4.26; github.com/MRCIEU/TwoSampleMR)¹⁴.”

Results line 313-323, page 12:

“Causal interference between the basal ganglia nuclei and common brain disorders

To investigate the potential causal links between basal ganglia nuclei and eight common brain diseases, Mendelian randomization was performed using the same GWAS summary statistics as for conjFDR: ADHD, ASD, BIP, MDD, SCZ, ALZ, MIG, and PD. Individual Mendelian randomization analyses of the disorders with each basal ganglia nuclei (Supplementary table 19 & 20) identified that the greater genetic liability to ALZ may cause lower accumbens volume (weighted-median: $\beta = -5.36$, $p_{adj} = 7.31 \times 10^{-3}$; weighted-mode: $\beta = -6.14$, $p_{adj} = 3.40 \times 10^{-3}$; MR-PRESSO: $\beta = -4.81$, $p_{adj} = 4.02 \times 10^{-2}$). Meanwhile, there was weak evidence that greater volume of the caudate (IVW: $\beta = 0.0011$, $p_{adj} = 6.74 \times 10^{-3}$; MR-PRESSO: $\beta = 1.10 \times 10^{-3}$, $p_{adj} = 1.31 \times 10^{-2}$) and putamen (Egger: $\beta = 0.0012$, $p_{adj} = 4.67 \times 10^{-2}$) may increase risk of PD.”

Discussion line 436-439, page 16:

“Through Mendelian randomization, two independent methods show that genetically determined larger caudate and putamen volume is associated with a slight increase in the risk of PD, supporting the LD-score regression results.”

Discussion line 444-453, page 16:

“Additionally, Mendelian randomization showed potential causal effect of ALZ on smaller nucleus accumbens volume and higher striatal volume on PD. The results indicating that genetic predisposition to ALZ influence nucleus accumbens volume is supported by multiple Mendelian randomization methods, increasing the robustness of the findings. In addition to the hallmark feature of medial temporal lobe atrophy, accumbens atrophy has also been found in ALZ patients, correlating with lower cognitive scores ¹⁵. As a limitation, individuals with ALZ and PD are included in the imaging sample, but the confounding effect is negligible as only five and 17 of the 34,794 individuals included in the current study are diagnosed with ALZ and PD, respectively (Supplementary figure 14).”

Adding behavioral and/or personality traits to the disease-centered analyses may further help understand the role of the basal ganglia in behavior and disease.

Response: We appreciate the reviewer's suggestion to include behavioral and/or personality traits in our analyses to gain insights into the role of the basal ganglia in behavior and disease. While we acknowledge the importance of considering these factors, our current focus is primarily on exploring the genetic underpinnings of basal ganglia volumes and their associations with neurological and psychiatric diseases.

Given the objectives of our study, the comprehensive inclusion of behavioral and personality traits is beyond its current scope. However, we recognize the potential value in examining these aspects in future investigations.

Reproducibility

The conditional analysis methods are briefly explained, but to me the rationale and methods remain in part unclear. What do the authors mean by “leverages cross-phenotype enrichment” as mentioned in the Methods?

Response: We appreciate the reviewer's comment regarding the conditional analysis methods and their rationale. We have revised the Methods section to provide a clearer explanation of the term 'leverages cross-phenotype enrichment.' In our updated description, we specify that the method estimates the posterior probability that a single nucleotide polymorphism (SNP) has no association with the primary trait, given that the p-values for that SNP in both the primary and conditional traits are as small as or smaller than the observed p-value. This process involves re-ranking the test statistics of the primary phenotype based on the strength of the association with the secondary phenotype, ultimately improving the detection of genetic variants associated with the primary trait. Moreover, we introduced the conjFDR approach at $FDR < 0.05$, which extends the conditional FDR. The conjFDR approach identifies common genetic loci between basal ganglia and the included brain disorders. It estimates the posterior probability that a SNP is null for either trait or both, given that the p-values for both phenotypes are as small as or smaller than the p-values for each trait individually. The method considers the highest of the two condFDR statistics for a specific SNP, providing an estimate of the posterior probability that a SNP is null for one or both characteristics.

We hope this clarification enhances the understanding of the methods employed in our study. Thank you for your valuable feedback.

Based on these methods, the authors state that “Conditional Q–Q plots conditioning the multivariate statistic of basal ganglia on the disorders and vice versa clearly demonstrated a pattern of pleiotropic enrichment in both directions (Supplementary Fig. 11)”. However, they do not explain what makes this pleiotropic enrichment so apparent in Supplementary Figure 11.

Response: We appreciate the reviewer's comment on the conditional Q-Q plots presented in Supplementary Fig. 13 (previously Supplementary Fig. 11) and the need for further explanation on what makes the pleiotropic enrichment apparent. In Supplementary Figure 13, the conditional Q-Q plots illustrate the relationship between the expected and observed significance of variants in the primary phenotype (basal ganglia or brain disorders) when these variants are stratified based on their p-values in the conditional phenotype. We have revised the figure text to provide a more detailed description. For more details, see methods papers and reviews ¹⁶⁻¹⁸.

Figure text to explain the pleiotropic enrichment:

Conditional Q-Q plots for basal ganglia on the brain disorders (left) and vice versa (right). The conditional Q-Q plot shows the relation between expected (x axis) and observed (y axis) significance of variants in the primary phenotype when variants are stratified by their p-values in the conditional phenotype. A sequence of four nested strata is presented: all SNPs (blue), $p_{\text{conditional_phenotype}} < 0.1$ (red), $p_{\text{conditional_phenotype}} < 0.01$ (orange) and $p_{\text{conditional_phenotype}} < 0.001$ (purple). Dashed black line demonstrates expected behaviour under no association. Observed significant leftward shift for the group of SNPs with higher significance indicated genetic enrichment and possible shared genetic background between basal ganglia and the brain disorders and vice versa.

Result section, line 276-280:

Conditional Q–Q plots conditioning the multivariate statistic of basal ganglia on the disorders and vice versa clearly demonstrated a pattern of pleiotropic enrichment in both directions (Supplementary Fig. 13) as a leftward shift in the group of SNPs with higher significance. This shift towards significance in the conditional Q-Q plot implies that the genetic variants associated with one phenotype (either basal ganglia or brain disorders) are also enriched for association with the other phenotype, supporting the notion of shared genetic factors or pleiotropy, as outlined ¹⁶⁻¹⁸.

Clarifications

Please clarify the statement on lines 156-160. Specifically, what do the authors define as a distributed genetic architecture?

Response: We appreciate the reviewer's request for clarification on the statement regarding a 'distributed genetic architecture' in our manuscript. In the context of our study, the term 'distributed genetic architecture' refers to the scenario when rather than having distinct genetic influences for each individual volume, there is a shared or distributed genetic influence that affects multiple basal ganglia volumes simultaneously, also named “polygenic pleiotropy”. The multivariate approach allows us to capture the joint genetic effects on multiple basal ganglia volumes collectively, providing evidence for a common or distributed genetic influence, as seen in other brain regions ¹⁹. Furthermore, this is further supported by the large overlap of loci between univariate GWASs of the different basal ganglia nuclei.

Additionally, we performed genetic correlation analysis of the individual basal ganglia volumes, and the results are provided in Supplementary Table 2. These analyses further

support the concept of a distributed genetic architecture by demonstrating the extent of shared genetic factors.

Minor comments

The sentence “It is, however, only directly significant association between genetic risk of PD and genetic architecture of basal ganglia” appears to be incorrect.

Response: We appreciate the comment by the reviewer. Upon revisiting the statement in question, we acknowledge that the sentence may benefit from clarification. We would like to highlight that the remark refers specifically to the directly significant association revealed through LD-score regression-based genetic correlations between the genetic risk of Parkinson's disease (PD) and the genetic architecture of basal ganglia. This observation is outlined in detail in Supplementary Fig. 12 and Supplementary Table 10.

The sentence is now changed to:

Discussion line 434-436, page 16:

“Notably, the sole significant correlation revealed through LD-score regression-based genetic correlations was a positive association between genetic risk of PD and genetic architecture of basal ganglia (Supplementary Fig. 12 and Supplementary Table 10).”

Scientific notation is in the form of 1e-5, where 1x10⁻⁵ would be preferred.

Response: We appreciate the reviewer's feedback regarding scientific notation. In response to this suggestion, we have revised the notation in our manuscript to align with the preference for expressing it as 1x10⁻⁵ rather than the abbreviated form 1e-5. We believe this adjustment enhances the clarity of the presentation and aligns with the conventional style.

The authors found 72 statistically significant loci. They state that 51 are novel. While they may indeed not have been related to subcortical volumes before, these loci are not novel in the sense that the multivariate trait has not been investigated before. I recommend choosing a different term or elaborating.

Response: We appreciate the reviewer's comment of the term 'novel' in the context of our study. We acknowledge that our multivariate basal ganglia GWAS represents the first exploration of its kind, providing a unique context for identifying loci compared to previous univariate GWASs focusing on individual volumes. While we understand the reviewer's perspective, we would like to emphasize that adopting a conservative approach and labeling loci identified in multivariate GWAS but previously found in univariate GWAS as 'non-novel' is a common practice in the field. We believe maintaining this distinction aids in transparent reporting. However, in line with the Reviewer's comment, we have now clarified the definition of novel applied in the manuscript.

Definition in results line 188-193, page 7:

“A multivariate GWAS deployed using the MOSTest framework⁸ identified 72 significant independent loci, including 50 novel loci (Supplementary Table 3). We here adopt a

conservative definition of novelty, wherein we characterize it as loci identified through the multivariate MOSTest approach for the basal ganglia as a unified functional unit, ensuring non-overlapping with any loci previously identified in univariate GWASs for any of the individual basal ganglia nuclei.”

References

- 1 Hibar, D. P. *et al.* Common genetic variants influence human subcortical brain structures. *Nature* **520**, 224-229 (2015).
- 2 Satizabal, C. L. *et al.* Genetic architecture of subcortical brain structures in 38,851 individuals. *Nature genetics* **51**, 1624-1636 (2019).
- 3 Bycroft, C. *et al.* The UK Biobank resource with deep phenotyping and genomic data. *Nature* **562**, 203-209 (2018). <https://doi.org/10.1038/s41586-018-0579-z>
- 4 Littlejohns, T. J. *et al.* The UK Biobank imaging enhancement of 100,000 participants: rationale, data collection, management and future directions. *Nature communications* **11**, 2624 (2020). <https://doi.org/10.1038/s41467-020-15948-9>
- 5 van der Meer, D. *et al.* Making the MOSTest of imaging genetics. *Biological psychiatry* **87**, S304-S305 (2020).
- 6 Freshour, S. L. *et al.* Integration of the Drug-Gene Interaction Database (DGldb 4.0) with open crowdsourcing efforts. *Nucleic acids research* **49**, D1144-d1151 (2021). <https://doi.org/10.1093/nar/gkaa1084>
- 7 Nelson, A. B. & Kreitzer, A. C. Reassessing models of basal ganglia function and dysfunction. *Annual review of neuroscience* **37**, 117-135 (2014). <https://doi.org/10.1146/annurev-neuro-071013-013916>
- 8 van der Meer, D. *et al.* Understanding the genetic determinants of the brain with MOSTest. *Nature communications* **11**, 1-9 (2020).
- 9 van der Meer, D. *et al.* The genetic architecture of human cortical folding. *Sci Adv* **7**, eabj9446 (2021). <https://doi.org/10.1126/sciadv.abj9446>
- 10 Bowden, J., Davey Smith, G., Haycock, P. C. & Burgess, S. Consistent estimation in Mendelian randomization with some invalid instruments using a weighted median estimator. *Genetic epidemiology* **40**, 304-314 (2016).
- 11 Bowden, J., Davey Smith, G. & Burgess, S. Mendelian randomization with invalid instruments: effect estimation and bias detection through Egger regression. *International journal of epidemiology* **44**, 512-525 (2015).
- 12 Relton, C. L. & Davey Smith, G. Two-step epigenetic Mendelian randomization: a strategy for establishing the causal role of epigenetic processes in pathways to disease. *International journal of epidemiology* **41**, 161-176 (2012).
- 13 Verbanck, M., Chen, C.-Y., Neale, B. & Do, R. Detection of widespread horizontal pleiotropy in causal relationships inferred from Mendelian randomization between complex traits and diseases. *Nature genetics* **50**, 693-698 (2018).
- 14 Hemani, G. *et al.* The MR-Base platform supports systematic causal inference across the human phenome. *eLife* **7**, e34408 (2018).
- 15 Nie, X. *et al.* Subregional structural alterations in hippocampus and nucleus accumbens correlate with the clinical impairment in patients with Alzheimer’s disease clinical spectrum: parallel combining volume and vertex-based approach. *Frontiers in neurology* **8**, 399 (2017).

- 16 Smeland, O. B., Frei, O., Dale, A. M. & Andreassen, O. A. The polygenic architecture of schizophrenia - rethinking pathogenesis and nosology. *Nature reviews. Neurology* **16**, 366-379 (2020). <https://doi.org:10.1038/s41582-020-0364-0>
- 17 Andreassen, O. A., Hindley, G. F. L., Frei, O. & Smeland, O. B. New insights from the last decade of research in psychiatric genetics: discoveries, challenges and clinical implications. *World Psychiatry* **22**, 4-24 (2023). <https://doi.org:10.1002/wps.21034>
- 18 Smeland, O. B. *et al.* Discovery of shared genomic loci using the conditional false discovery rate approach. *Human genetics* **139**, 85-94 (2020).
- 19 Bahrami, S. *et al.* Distributed genetic architecture across the hippocampal formation implies common neuropathology across brain disorders. *Nature communications* **13**, 1-9 (2022).

REVIEWERS' COMMENTS

Reviewer #1 (Remarks to the Author):

The authors are commended for meticulously addressing all of my comments and suggestions

Reviewer #2 (Remarks to the Author):

The authors made thorough revisions that resulted in a much-improved manuscript. I would like to thank the authors for what clearly has been a lot of work. The responses to my comments were comprehensive and convincing. The added value over existing literature is more apparent now.

In particular, the authors clarified well how, despite having a similar sample size to an earlier study, more power and new findings were achieved thanks to a more homogeneous cohort, and methodological advances.

I appreciate that authors pro-actively included Mendelian randomization analyses.

I have only one concern based on the newly added Mendelian randomization analyses. The authors found a potential link between genetic liability to ALZ and lower nucleus accumbens volume. They rightfully discussed a potential confounding effect due to ALZ patients being included in the imaging cohort. They addressed this as follows:

"As a limitation, individuals with ALZ and PD are included in the imaging sample, but the confounding effect is negligible as only five and 17 of the 34,794 individuals included in the current study are diagnosed with ALZ and PD, respectively (Supplementary Fig. 14)."

The fact that only a few ALZ (and PD) cases are included in the cohort could underestimate the influence of confounding. Specifically, persons with prodromal symptoms, but no ALZ diagnosis (yet) could be present. For obvious reasons these persons cannot all be excluded from the analysis. However, a sensitivity analysis including young individuals would better tackle this issue. Furthermore, it would give more insight in whether nucleus accumbens volumes are aberrant in persons at risk of ALZ long before they develop ALZ.

Response to all reviewers

We would like to thank the reviewers for their constructive feedback that clearly helped improving our manuscript

Reviewer response letter:

REVIEWER # 1:

The authors are commended for meticulously addressing all of my comments and suggestions

Response: We would like to extend our heartfelt appreciation to the reviewer for the commendable feedback and acknowledgement of our efforts in addressing the comments and suggestions from the previous rounds of revision. The guidance has been instrumental in refining and enhancing the quality of our work.

REVIEWER # 2:

The authors made thorough revisions that resulted in a much-improved manuscript. I would like to thank the authors for what clearly has been a lot of work. The responses to my comments were comprehensive and convincing. The added value over existing literature is more apparent now.

In particular, the authors clarified well how, despite having a similar sample size to an earlier study, more power and new findings were achieved thanks to a more homogeneous cohort, and methodological advances.

I appreciate that authors pro-actively included Mendelian randomization analyses.

Response:

Thank you for acknowledging the improvements in our revised manuscript. We're glad our responses and the updated analyses, including the Mendelian randomization, have met with your approval. Your feedback has been invaluable in enhancing our study's quality and relevance.

I have only one concern based on the newly added Mendelian randomization analyses. The authors found a potential link between genetic liability to ALZ and lower nucleus accumbens volume. They rightfully discussed a potential confounding effect due to ALZ patients being included in the imaging cohort. They addressed this as follows:

“As a limitation, individuals with ALZ and PD are included in the imaging sample, but the confounding effect is negligible as only five and 17 of the 34,794 individuals included in the current study are diagnosed with ALZ and PD, respectively (Supplementary Fig. 14).”

The fact that only a few ALZ (and PD) cases are included in the cohort could underestimate the influence of confounding. Specifically, persons with prodromal symptoms, but no ALZ diagnosis (yet) could be present. For obvious reasons these persons cannot all be excluded from the analysis. However, a sensitivity analysis including young individuals would better tackle this issue. Furthermore, it would give more insight in whether nucleus accumbens volumes are aberrant in persons at risk of ALZ long before they develop ALZ.:

Response: We appreciate the reviewer's suggestion to perform a sensitivity analysis including younger individuals to explore whether nucleus accumbens volumes are affected in individuals at risk of ALZ before clinical manifestation, but after onset of neuropathology. The presence of preclinical pathology is indeed a relevant concern when assessing the potential etiology of neurodegenerative disorders.

The reviewer correctly points out that including younger individuals might help to exclude early structural brain changes in prodromal ALZ or prodromal PD. However, reducing the age cutoff to 65 would result in the exclusion of a substantial portion of the cohort, with a consequently significant reduction in the statistical power. The length and time of events in the prodromal phases are largely understudied, we have operated with a certain longitudinally determined age cut-off that best balances sensitivity and specificity given these limitations.

Additionally, the regional specificity of preclinical neuropathology should be considered. The neuropathological cascade of ALZ begins in the neocortex and allocortex and only later extends to the basal ganglia, and in PD, the involvement of the basal ganglia follows much initial pathology in the olfactory bulb and lower brain stem. Accordingly, there are no neuropathological support to univariate accumbens, putamen or caudate GWASs being confounded by preclinical neuropathology affecting these regions.

However, in response to the reviewer's comments on conducting a sensitivity analysis with a younger cohort, we have re-analyzed our Mendelian randomization results with two alternative age cut-offs: 75 and 65 yrs. Whilst the former age group showed a slight reduction in power (n=32,455), the latter experienced a much more pronounced decrease (n=17,591). Importantly, the sensitivity analyses excluding all participants 75 years old and older substantiate our already reported results, which demonstrate consistent associations across methods for both PD and ALZ in this age subsets. This includes a significant positive association between genetic markers correlating with higher putamen and caudate volumes and PD risk, as well as a significant association between ALZ genetic risk and lower accumbens volume. For the age subset of participant younger than 65 years old the genetic correlation between higher striatal volume and PD risk was again replicated, while the association between ALZ risk and lower accumbens volume was not significant. Whether this is due to this age subset cutting the cohort in half, or preclinical ALZ patients indeed confounding our results, is uncertain. In our revised manuscript, however, we suggest to add this uncertainty:

In summary, our sensitivity analyses, which span two different age cut-offs, suggests that the genetic risk of PD may be associated with increased putamen and caudate volumes, and that the genetic risk of ALZ may be associated with decreased accumbens volume, albeit in a nuanced manner when considering different age groups. The attested attenuation of association signals in the youngest cohort, especially for the ALZ-related outcome (accumbens volume), may align with the presumption of lesser confounding by preclinical disease in younger individuals. Conversely, the robustness of the PD-related findings in participants younger than 65 hints at a genuine link less likely to be confounded by incidental pathology. It is also warranted to reiterate that the basal ganglia are not the earliest regions affected in the pathogenesis of neither PD nor ALZ, as evidenced by established staging systems for ALZ (Thal phase and Braak NFT stage) and PD (Braak LB stage). Given this pathology, it is plausible to consider that volumes of nucleus accumbens, putamen, and caudate could remain relatively unaffected even in the presence of preclinical pathology.

For clarity, the main changes to the manuscript is added below. We marked all changes in the manuscript related to the reviewer's comment with **green** colour.

Results section from line 297, page 12:

“As both ALZ and PD exhibit a long preclinical disease trajectory, sensitivity analyses were performed using two alternative age cut-offs: 75 and 65 years old, encompassing n=32,455 and n=17,591 individuals, respectively. The results remained robust when using the 75-year age cut-off. However, for the 65-year age cut-off, only the association between greater volume of caudate (IVW: beta= 0.0011, $p_{adj}= 1.43 \times 10^{-2}$; weighted-median: beta=0.0014, $p_{adj}=1.06 \times 10^{-2}$) and the risk of PD remained significant after adjustment for multiple testing (Supplementary Table 21).”

Discussion section from line 408 page 17:

“While implementing sensitivity analysis (Supplementary Table 21) with stringent age filters, we observed an anticipated decline in statistical power; nevertheless, the persistent association between higher striatal volumes and PD genetic markers underscores the potential linkage, as detailed in the supplementary material. Despite the diminished cohort size, the consistency of our PD-related results across age brackets bolsters their validity. Conversely, the non-significant correlation between accumbens volume and AD risk in participants under 65 suggests that our findings may be influenced by preclinical neuropathology affecting this result, challenging the notion of genetic AD risk as a stand-alone causal factor for reduced accumbens volume.”

Methods section from line 983 page 35:

“Furthermore, we performed a sensitivity analysis to assess the robustness of our Mendelian randomization findings. This analysis involved young individuals with two alternative age cut-offs: 75 and 65 years old, including n=32,455 and n=17,591 individuals, respectively.”